

# Drone-based photogrammetry combined with deep-learning to estimate hail size distributions and melting of hail on the ground

Martin Lainer[1], Killian P. Brennan[1,a], Alessandro Hering[1], Jérôme Kopp[2], Samuel Monhart[1], Daniel Wolfensberger[1], and Urs Germann[1]

[1]Federal Office of Meteorology and Climatology, MeteoSwiss, Locarno-Monti, Switzerland
[a]now at: Institute for Atmospheric and Climate Science, ETH Zurich, Zurich, 8092, Switzerland
[2]Oeschger Centre for Climate Change Research and Institute of Geography, University of Bern,, Switzerland

**Correspondence:** Martin Lainer (martin.lainer@meteoswiss.ch)

**Abstract.** Hail is a major threat associated with severe thunderstorms and an estimation of the hail size is important for issuing warnings to the public. Operational radar products exist that estimate the size of the expected hail. For the verification of such products, ground based observations are necessary. Automatic hail sensors, as for example within the Swiss hail network, record the kinetic energy of hailstones and can estimate with this the hail diameters. However, due to the small size of the

observational area of these sensors $(0.2\,\mathrm{m}^2)$ the estimation of the hail size distribution (HSD) can have large uncertainties. To overcome this issue, we combine drone-based aerial photogrammetry with a state-of-the-art custom trained deep-learning object detection model to identify hailstones in the images and estimate the HSD in a final step. This approach is applied to photogrammetric image data of hail on the ground from a supercell storm, that crossed central Switzerland from southwest to northeast in the afternoon of June 20, 2021. The hail swath of this intense right-moving supercell was intercepted a few minutes

after the passage at a soccer field near Entlebuch (Canton Lucerne, Switzerland) and aerial images of the hail on the ground were taken by a commercial DJI drone, equipped with a $50$ megapixels full frame camera system. The average ground sampling distance (GSD) that could be reached was $1.5\,\mathrm{mm}$ per pixel, which is set by the mounted camera objective with a focal length of $35\,\mathrm{mm}$ and a flight altitude of $12\,\mathrm{m}$ above ground. A 2D orthomosaic model of the survey area $(750\,\mathrm{m}^2)$ is created based on 116 captured images during the first drone mapping flight. Hail is then detected by using a region-based Convolutional

Neural Network (Mask R-CNN). We first characterize the hail sizes based on the individual hail segmentation masks resulting from the model detections and investigate the performance by using manual hail annotations by experts to generate validation and test data sets. The final HSD, composed of 18209 hailstones, is compared with nearby automatic hail sensor observations, the operational weather radar based hail product MESHS (Maximum Expected Severe Hail Size) and some crowdsourced hail reports. Based on the retrieved drone hail data set, a statistical assessment of sampling errors of hail sensors is carried out.

Furthermore, five repetitions of the drone-based photogrammetry mission within about $18\,\mathrm{min}$ give the unique opportunity to investigate the hail melting process on the ground for this specific supercell hailstorm and location.



## 1 Introduction

Hail is a severe danger associated with thunderstorms and its impact increases with its size. Therefore, the estimation of the hail size is important for an appropriate warning of the public and to asses the damage. The hailstorms in the period between 18 June

and 31 July 2021 were extremely intense over Switzerland (Kopp et al., 2022). According to la Mobilière (2021), the amount of storm-related losses in Switzerland accumulated to around 340 million (CHF) in the month of June alone and large hail played a significant role. Operational weather radar-based algorithms exist, which try to compute the maximum expected severe hail size (MESHS, Treloar, 1998) and probability of hail (PoH, Waldvogel et al., 1979) within a thunderstorm. In Switzerland, those products are derived from five C-band weather radars operating in the complex terrain of the alps (Germann et al., 2022)

and have a spatial resolution of $1\,\mathrm{km}^2$. For the verification and improvements of radar-based hail products, ground-based observations are needed.

Beside traditional hailpads, which are cost effective and do not provide any time information, promising other observational approaches and tools include newly developed automatic hail sensors (Löffler-Mang et al., 2011) and crowdsourced hail reports (Barras et al., 2019). Recently, a network of 80 automatic hail sensors was installed in the three most hail-prone regions of

Switzerland (Jura, southern Ticino and Napf) according to climatology studies (Nisi et al., 2018, 2016) in the framework of «The Swiss Hail Network» project (Romppainen-Martius, 2022; Kopp et al., 2022). These sensors record the kinetic energy of single hailstones and infer their sizes with a precise time information, but provide no shape (axis ratio) information and have only a small impact area of $0.2\,\mathrm{m}^2$. The crowdsourced hail data usually only gives information about the largest observed hail diameters. The quality control of crowdsourced hail observations is difficult and the data has a low accuracy due to predefined

fixed size categories (smaller than coffee bean: $> 0$–$5\,\mathrm{mm}$, coffee bean: $5$–$8\,\mathrm{mm}$, 1 CHF coin: $23\,\mathrm{mm}$, 5 CHF coin: $32\,\mathrm{mm}$, golf ball: $43\,\mathrm{mm}$, tennis ball: $68\,\mathrm{mm}$). Another source of uncertainty arises from the size estimates submitted by the app users (Barras et al., 2019).

Hail sensors cannot capture the entire hail size distribution (HSD) of a hail storm due to their limited area (Kopp et al., 2023) and crowdsourced reports only give information about the largest hailstone, therefore they cannot be used to infer a complete

HSD. A new technique, called *HailPixel*, has been introduced by Soderholm et al. (2020) for measuring the size distribution of hail over a large detection area using aerial imagery captured from a small unmanned aircraft and deep-learning combined with computer vision feature extraction. They show exemplary results from a *HailPixel* survey of a hailstorm in San Rafael (Argentina) in the context of existing studies and point out potential improvements for future hail surveys. With a sample of 15983 measured hailstone sizes, they were able to precisely define the shape and tails of the HSD.

A main advantage of image-based hail data sets over crowdsourced, hail sensor and hailpad data is, that the larger detection area allows to derive a more complete HSD. Likewise with hailpads, the shape factor in the image plane can be easily determined as well. It is known, that hailstones usually have an oblate spheroid shape with mean axis ratios close to $0.8$, though they can sometimes have large protuberances (Knight, 1986). In this study, we present detailed statistics of the observed hail aspect ratios for a particular hail event and location.





On 20 June 2021, the ingredients for long-living and well-organized severe thunderstorms came together over Switzerland. An air mass with steep lapse rates from the southwest was advected on top of decent, low-level moisture. As the 12 UTC sounding from the meteorological station Payerne in Fig. 1 points out, during the course of the day SB (Surface-Based) CAPE (Convective Available Potential Energy) values of more than $2000 \, \mathrm{J \, kg^{-1}}$ developed in conjunction with a high 0–6 km wind shear of about $30 \, \mathrm{m \, s^{-1}}$. A hail producing strong supercell was chased on that day. The temporal track of this supercell from

radar data is shown in Fig. 2(a). At 12 UTC, when the radio sounding took place, the cell was located above the French alps (magenta circle). It entered Switzerland 30 min later and crossed the country in a period of 5 hours (s. storm track in Fig. 2(a)). A deeper analysis of the storm motion and storm relative winds based on the hodograph display (s. Fig. 1), shows that the environment favored the development of classical right-moving supercells (Houze et al., 1993) on that day.

Based on the MESHS products from the Swiss C-band radar network (Germann et al., 2022), we find that the hail swath

was nearly uninterrupted from the north-eastern edge of lake Geneva to the north-western edge of lake Zurich over a length of about 155 km. The maximal width of the hail swath part exceeding a MESHS value of 6 cm was in the order of 10 km. Figure 2(b) illustrates the radar derived MESHS signature from the supercell when it crossed central Switzerland, just a little south of the Napf region (predominantly rural area), where a cluster of automatic hail sensors are installed. MESHS reached 6.3 cm when the hail from the supercell was intercepted at a soccer field (magenta cross) near Entlebuch (Canton Lucerne). The same

MESHS values were present at the two closest hail sensor locations (HS1 and HS2 on map in Fig. 2(c) and (d)). At the two other sensor locations (HS3 and HS4) MESHS reached lower values of 48 mm and 58 mm (Fig. 2(d)). Interestingly, the very closest sensor HS1 (300 m SSW from the soccer field) did not get any impact during the passage of the hail core. Because of the absence of any record on this closest sensor, we use the hail data from the other 3 sensors, which are in the vicinity of the drone hail survey area. As a note, HS2 and HS4 are at a distance of 770 m, respectively 1470 m, to the NNE direction of the

soccer field, while HS3 is at a distance of 1150 m to the SSW (Fig. 2(c)).

Soderholm et al. (2020) provided recommendations for drone-based hail surveys in general: Ideally conduct them on uniform and contrasting backgrounds (cut or grazed turf grasses); Increase the camera resolution for capturing smaller hailstones; Minimize the melting of hailstones and avoid aerial surveys in areas with flowing water and conduct surveys as immediately as possible after hail fall stops to prevent further melting. Following those suggestion, we have put effort in using a camera

equipment that allows to take pictures in a higher resolution to increase the ground sampling distance (GSD) to $1.5 \, \mathrm{mm \, px^{-1}}$. Therefore, we are able to classify hailstones down into the small 3–6 mm bin size, compared to the minimum size of 2 cm in Soderholm et al. (2020). The drone-based hail survey in this study was performed on a soccer field with a visually homogeneous background, and an excellent drainage of water. A main difference to the approach of Soderholm et al. (2020) is the technical setup to estimate the size of the identified hailstones. Instead of using an additional computer-vision-based method, we here

use only the data from the deep-learning algorithm to directly estimate the hail sizes and shapes. In addition, we present an approach to address the major problem of the drone-based method, which is the melting of hailstones on the ground before the aerial images are captured. The idea to tackle this drawback is to introduce several identical, consecutive drone flight missions above the same area to capture the shrinking hailstone sizes and derive the rate of melting. This will allow, to a certain stage, to





approximate the expected largest hail sizes dating back to the start of the hail fall, if the exact times of the storm passage and
images are known.

In Sect. 2 the end-to-end chain is presented, starting from the data collection procedure, the equipment used and details about
the image data acquisition, followed by the post-processing and the automated task of hail object detection with deep-learning
algorithms and the final retrieval of the hail size distribution. The results and performed investigations are described in Sect. 3.
Further discussions to set the findings in a broader context are presented in Sect. 4. Conclusions, ideas and suggestions for
future analyses are given in Sect. 5.

## 2  Data and methods

Here we first go into the challenging part of the hail image data collection process (Sect. 2.1) and then provide an overview of
the technical devices (drone, camera system and aerial mapping strategy) used (Sect. 2.2). An introduction to the deep-learning
(Convolutional Neural Network) based method (Detectron2 framework, Wu et al. (2019)) to automatically detect hail objects
in the processed image data record is presented in Sect. 2.3. The hail detection and size estimation approach is evaluated based
on a validation and a test data set. Hereby we follow the *HailPixel* procedure described in Soderholm et al. (2020) with some
slight adaption that are mentioned briefly below. The *HailPixel* technique applies a two-stage approach, combining machine
learning for finding the center hail pixel and computer vision (CV) for an exact hail edge detection on the image lightness in
the HSL (Hue, Saturation, Lightness) color space. During a preliminary test phase, the two-stage approach from Soderholm
et al. (2020) was compared to a one-stage method using solely a deep-learning instance segmentation model based on Mask
R-CNN. It was found, that the size estimation based on our one-stage method led to better results for this specific hail case
than the two-stage approach.

### 2.1  Data collection and the experience from chasing hailstorms

A major challenge of drone based hail photogrammetry is the collection of data. Therefore we here briefly describe our strategy
to prepare the data collection process. Hail producing thunderstorms are highly localized phenomena and falling hail melt
quickly on the surface due to high (summer) air and soil temperature and sometimes strong rainfall following directly after
the hail. Thus, to encounter a thunderstorm, the drone operators need to be on site before the arrival of the storm. Therefore,
the availability of good nowcasting products and experienced interpretation are highly important. Aside the meteorological
challenges, the practical difficulties are even more pronounced. To obtain best possible quality of aerial images, we focused
on places where we were confident to encounter fresh cut meadows. Public soccer fields turned out to be most promising
target locations, which can be easily identified in interactive maps while being on the road, e.g. on https://map.geo.admin.ch/
(SwissGeoportal, 2023). In addition, major parts of the hail prone areas were scouted in advance to determine potential locations
to intersect a specific storm cell and familiarize with the local traffic routes.

For the preparation of the data collection process, different numerical weather prediction outputs have been consulted. We
aimed at situations where the NWP models showed supercell favoring conditions, i.e. high moisture content, favorable wind



shear conditions, degree of instability (CAPE) and the potential for triggering by meteorological forcing or by the terrain (s. for instance Holton (2004) for details about storm dynamics and the prediction of convective storms). During days with such favoring conditions, the drone operators were on standby in central Switzerland already in the morning hours to be ready to head towards potential regions of thunderstorm occurrence. Another valuable source are the forecasts by ESTOFEX (European

Storm Forecast Experiment, Groenemeijer et al. (2007)). Our experience has shown, that at least a level 2 on the ESTOFEX internal scale needs to be issued to have a realistic chance to interfere a hail producing cell. In general, the forecasts and evaluation of the synoptic situation across Europe provided on their website are highly valuable for the preparation process and determining, whether meteorological conditions are favorable on the next day.

On the day of an event, different nowcasting and observational products were used. Most importantly, the operational radar
images produced by MeteoSwiss served as a baseline to identify storms and nowcast the upcoming minutes to hours. Hereby, the 3-dimensional reflectivity information is crucial to not only identify the cell itself but to further estimate the strength and exact location of a potential hail core. Within the operational radar products, POH and MESHS was used. Our experience has shown that for promising results, POH needs to be $100\,\%$ and MESHS should reach stable values above $2\,\mathrm{cm}$. Furthermore, satellite images and lightning information, e.g. lightning jumps (Schultz et al., 2009; Chronis et al., 2015; Nisi et al., 2020),
help to focus on intensifying regions within the developing storm cells. Finally, real-time hail reports from the public can give a hint about the size of the hail that can be expected and to fine tune the final decisions for a successful hail core punch.

Following this strategy and using the tools mentioned, two drone-based hail photogrammetry surveys could be performed during five event days in 2021. In this study we present an analysis of the data collected on the 2021-06-20 to demonstrate the methodology. The data from the second available event can not be taken into account because of low quality of the data. In
particular, both, the light conditions and the background (longer grass on the soccer field) were not optimal and thus the data can unfortunately not be used for an in depth analyses.

We further want to mention as a disclaimer that for a successful hail core punch it is crucial to react fast on the basis of the instantaneous knowledge of the meteorological situation, but at the same time quick decisions can lead to a miss of the hail core. Therefore hail chasers should be resilient, have lot of patience and enthusiasm to catch the perfect storm and keep in
mind that storm chasing involves potential severe risks, including injury from large hail and extreme winds. In addition, traffic conditions can be dangerous due to heavy precipitation and strong winds. Safety should be the priority during storm chasing experiments. Preparation of potential shelter spaces and escape routes is recommended in case of worst-case scenarios.

## 2.2  Drone operation and image processing

The aerial hail photogrammetry missions were performed with a DJI Matrice 300 RTK drone equipped with a Zenmuse
P1 camera system, that has a full-frame sensor ($45\,\mathrm{MP}$) stabilized by a 3-axis gimbal and a focal length of $35\,\mathrm{mm}$. The synchronization of the camera, flight controller, RTK (Real Time Kinematic) GPS module, and gimbal takes place at the microsecond level and ensures a high accuracy of the image data.

Studies such as Guidi et al. (2020) or Fawcett et al. (2019) showed that a frontal and side image overlap between $70\,\%$ and $80\,\%$ is within an optimal range to produce an orthomosaic. An orthomosaic is a photogrammetrically orthorectified image



product that has been mosaicked from an image collection, correcting for geometric distortion and color matching the image data to create a seamless mosaic data set. The large overlap and the image redundancy usually allows for an efficient elimination of erroneous matches, which in turn improves the reliability of the 3-dimensional point cloud (3D model based on collections of individual points plotted in 3D space). For our performed flight missions an image overlap of $70\%$ for both sides (frontal and sideways) was applied.

The automated flight with the DJI Matrice 300RTK drone was planned from the DJI Pilot 2 application as a lawnmower (boustrophedonic) flight path without cross-hatch. Due to restrictions of the DJI control software, the minimal possible flight altitude was $12\,\mathrm{m}$ above ground. As demonstrated by Soderholm et al. (2020) or Bemis et al. (2014), a slow horizontal flight speed reduces the motion blur. A flight speed of $1\,\mathrm{m\,s^{-1}}$ was programmed to keep the motion blur within one image pixel. The image processing to produce a high resolution georeferenced orthorectified image (hereafter called simply orthophoto)

from the survey area was done with the open source software OpenDroneMap (ODM, OpenDroneMap (2020)). It is able to turn simple 2-dimensional images into: classified point clouds, 3-dimensional textured models, georeferenced orthorectified imagery or georeferenced digital elevation models. ODM makes use of OpenSfM (mapillary, 2020), which is a structure from motion (SfM) library written in Python on top of OpenCV (Bradski, 2000). The library serves as a processing pipeline for reconstructing camera poses and 3-dimensional scenes from multiple images. Here we make use of some basic modules for

SfM: Feature detection, feature matching, minimal solvers.

   The orthophoto construction can be broken down into the following main steps:

- Identification of matching points between the images.

- Reconstruction of the camera perspective and the position of each image for quality check and subsequent computation of the 3-dimensional coordinates of the matching points.

- Derivation of a DEM (digital elevation model) by using a reduced point cloud in 3-dimensional space.

- Construction of the orthophoto of the survey area by applying the DEM to spatially project every image pixel.

   The first flight mission after the passage of the hail core of the supercell started 14:37:28 (UTC), which is about $9.5\,\mathrm{min}$ after the start of the hail fall. Within $3{:}51\,\mathrm{min}$ a total of 116 images were taken at a constant altitude of $12\,\mathrm{m}$ above ground level. Each image has $8192 \times 5460$ pixels and was captured at $35\,\mathrm{mm}$ focal length with a manual set exposure time of $1/1000\,\mathrm{s}$, an

aperture of $f/5.6$ and ISO-25600 for the applied gain by the camera sensor. The resulting GSD of $1.5\,\mathrm{mm\,px^{-1}}$ is good enough to visually detect hailstones $> 5\,\mathrm{mm}$ and thus also the smallest size classification of hail (World Meterological Association, 2017).

   The quality report of the ODM processing of the 116 survey images revealed the reconstruction of a total of $14.916.215$ dense points and a mean GPS error of $0.34\,\mathrm{m}$. The total hail area coverage of the ortophoto within the soccer field as presented

in Fig. 3(a) is $750\,\mathrm{m^2}$. By inspecting the computed digital terrain model of the area, a maximal change in elevation of $0.5\,\mathrm{m}$ is found. For an independent verification of the GSD, reference objects (s. Fig. 3(b)) were placed on the soccer field before the drone image capturing started. Those objects were laminated printouts of geometric shapes in black and white, e.g. circles





of diameter $10\,\text{mm}$ and squares with side lengths of $75\,\text{mm}$. Cross checking of the $10\,\text{mm}$ white circles yielded a diameter between 6 and 7 pixels, equivalent to the metric range 9–10.5 mm. Due to a slight overexposure in combination with the motion blur, the black circles on white background appeared much smaller than the white ones on the black background.

## 2.3 Object detection and size estimation

Object detection is a technology related to computer vision and image processing that tries to detect instances of semantic objects of a certain class (e.g. cats, dogs, cars, buildings, etc.) in digital images and videos. Generally, the methods for object detection fall into either neural network-based or non-neural network-based approaches. A good overview about the techniques and developments in object detection over the last two decades is shown in the road map of milestones in object detection by Fig. 2 in Zou et al. (2019). In recent years, many of the latest available neural network detection engines (e.g. AlexNet, VGG, GoogleNet, ResNet, DenseNet) have been applied to object detection. For example, the Mask R-CNN (He et al., 2020), as one of the state-of-the-art models for instance object segmentation tasks, uses the ResNet (He et al., 2016) detection engine. This residual learning framework was designed to simplify the training of substantially deep neural networks.

We used the deep-learning toolbox Detectron2 Wu et al. (2019) as a starting point to train a model for visual hail recognition. Its flexible design makes it easy to switch between different tasks such as object detection, instance segmentation or panoptic segmentation. It has built-in support for popular data sets like the MS COCO (Microsoft Common Objects in Context) described in Lin et al. (2014) and many backbone combinations of Faster/Mask R-CNN: ResNet (Residual Neural Network) in combination with FPN (Feature Pyramid Network), C4 (Convolution 4) as single scale feature map, or dilated convolution. Further, Detectron2 provides ready-to-use baselines with pre-trained model weights. One of those pre-trained model weights on the MS COCO data set is used to train a new model on a custom designed hail class. It should be mentioned, that the training on one class always requires at least a second class, namely the image background. Ideally for the model to be as general as possible in the detection of hail, it would be necessary to train against various background types. Obviously, with only one captured hail event trained on a certain background (e.g. soccer field) the model performance will change (likely decrease) on a different background.

### 2.3.1 Image data preparation

The ODM software allows to directly export the originally produced GeoTIFF format of the orthophoto into an uncompressed PNG (Portable Network Graphics) image format. For further processing only the PNG version of the orthophoto has been used. The large file of width $24500\,\text{px}$ and height $22000\,\text{px}$ has a storage size of about $2\,\text{GB}$ and a total number of $5.39 \times 10^8$ pixels. As shown in Fig. 3(a) the orthophoto does not cover the full rectangular area in the image, thus the area that will be analyzed by the object detection algorithm consists only of approximate $5 \times 10^8\,\text{px}$. Given the GSD of $1.5\,\text{mm}\,\text{px}^{-1}$, the area size reaches $750.4\,\text{m}^2$.

The high demand of computational resources during the training of the Mask R-CNN makes it unavoidable to work with smaller image tile files. Several tests have shown that an image tile size of $500 \times 500\,\text{px}$ is a reasonable compromise. Therefore the orthophoto was divided into 2156 PNG image tiles. Later, a random selection of those image tiles was applied to assign





**Table 1.** Overview of the performed variations of the 3 hyper-parameters: Learning rate (LR, 1st row), $\gamma$ value (2nd row) and batch size (BS) per image (3rd row). The hyper-parameter combination of the model with the lowest validation loss after 3000 training iterations are highlighted in red.

| LR | **0.0001** | 0.00025 | 0.0005 | 0.001 |
|---|---|---|---|---|
| $\gamma$ | 0.1 | | **0.5** | |
| BS | 128 | | **256** | |

tile images to train (150 images), validate (33 images) and test (33 images) the model (s. Fig. 3(c)). The idea behind is to use 10 % of all available image tiles as reference data. This yields a total of 216 images and those were again divided into 70 % for training and the remaining bulk equally split into 15 % each for the validation and test data. These three data sets of images are further processed with the Computer Vision and Annotation Tool (CVAT, Sekachev et al. (2020)) to manually annotate all

clearly visible hailstones and to export the final annotation data set. CVAT supports multiple annotation formats, including the COCO format that is a good choice for the Detectron2 framework. The annotation files are JSON (JavaScript Object Notation) based and store information about each image tile path, width, height, annotation identifiers of the hailstones and the polygon coordinates defining their binary masks. Overall a total of 937 hailstone annotations are contained in the training set, 249 in the validation set and 215 in the test set. In these annotation data sets all hailstones were visually identified by a human expert A

and the identified hailstones were used as annotations during the training and validation of the neural network. To account for differences in the visually determined annotations, tow more experts (B and C) annotated the test data set consisting of 33 tile images. Thus the test data set is created by three independent experts and not used during the training and validation process nor affects the hyper-parameter adjustments and can be used as a good benchmark against the CNN results.

### 2.3.2   Hail detection and size estimation - training, validation and testing

A NVIDIA GeForce RTX$^{\text{TM}}$ 3060 Ti was used to efficiently train the Mask R-CNN model on the custom hail data set. This GPU model has 4864 CUDA (Compute Unified Device Architecture) cores and in total 8 GB GDDR6 RAM available. A default configuration of Detectron2 is used for a first estimate of the hyper-parameter tuning. For the hail detection training the default backbone network (ResNet) was applied and the pre-trained model on the MS COCO data set used a Resnet and FPN combination. The large MS COCO data set consists of about $2 \times 10^5$ annotated images with a total of 80 different object

classes and it is thus an ideal starting point to train deep-learning models to recognize, label, and describe objects.

A set of 16 training model runs («run-0» to «run-15») going through several standard hyper-parameter combinations (s. Table 1) were conducted on the GPU device to find the most suitable trained model. Each run consisted of 3000 training iterations. With the use of 1 GPU loaded with two images per batch and 150 training images in total, 75 training iterations are needed for one epoch time. Thus 3000 iterations translate into 40 epoch times for each of our performed runs. The number

of epochs needed highly depends on the diversity of the data, and as ours consists only of one object class, the chosen 40 epochs are enough.





The internal model evaluation period was set to 100 iterations. This means, that there are 30 available points along the iterations where the performance of the model is automatically back-tested against the validation data set. Figure 4 compares the progress of total and validation loss for the 16 performed training runs. The thick bold lines show the run which had the lowest validation loss after 40 training epochs. To chose the best model, we performed a more detailed evaluation of the model runs by means of commonly used metrics in object detection. The accuracy of an object recognition model depends on the quality and number of training regions, the input image data, the model parameters, and the accuracy requirement threshold. Usually, the IoU (Intersection over Union) ratio is used as a threshold to determine whether a predicted result is a true positive ($TP$) or a false positive ($FP$). The IoU ratio is usually the overlap between the surrounding rectangle around a predicted object and the surrounding rectangle around the same object in the reference annotation data set. In this study, we use the IoU retrieved from the binary mask areas and not from the surrounding rectangles. Following the standard COCO evaluation procedure, the set of IoU ratios ranges from 0.5 to 0.95 in steps of 0.05. The minimum IoU value for a matching detection ($TP$ result) is 0.5.

The normal procedure when training a deep-learning model is to split the reference annotation data into a train and a test set. Because we want the test data set to be locked down until we are confident enough about our trained model, we do another division and split a validation set out of the train set. In this scenario we end up with three data sets. Usually we want to compare how well the model is performing on the validation set during the training, in order to know when are we at risk of over fitting the model to the training data. In the end, the final evaluation of the model performance should be computed on the test data set, as the model training was totally independent from it. Further we used the test set to investigate the discrepancies between three professional experts, who annotated the hailstones in those images (s. Fig. 7).

The model «run-3» was selected for the final hail detection and size estimation. Every single tile image was pushed through this Mask R-CNN model version («run-3») and the binary masks of all found hail objects were saved in separate Python structures linked to the individual images. Regarding the full orthophoto area (Fig. 3(a)), split into the 2156 tile images, the Mask R-CNN model classified 18209 objects as hailstones. A few large objects (e.g. leaves) were wrongly classified as hail and manually removed to guarantee a correct representation of the largest hail size bins in the distribution.

In pattern recognition, information retrieval, object detection and classification (machine learning), precision and recall (Eq. (1) and Eq. (2)) are standard performance metrics (Powers, 2020) that apply to data retrieved from a collection or sample space. Precision $[0,1]$ is a measure of result relevancy, while recall $[0,1]$ is a measure of how many truly relevant results are returned. A model system with high recall but low precision returns many identified objects, but most of these objects are incorrectly labeled (False positive) when compared to the validation labels. On the other hand, high precision but low recall is just the opposite, where only few objects are identified and most them are labeled correct, when compared to the validation labels. An ideal system with high precision and high recall will return a realistic amount of positive results. The $F1$ score in Eq. (3) combines precision and recall metrics into one unified measure and is designed to handle imbalanced data effectively.



$$Precision = \frac{TP}{TP + FP} \tag{1}$$

$$Recall = \frac{TP}{TP + FN} \tag{2}$$

$$F1 = 2 \cdot \frac{Precision \cdot Recall}{Precision + Recall} \tag{3}$$

With regard to Fig. 5, a reasonable compromise between high precision and high recall values for «run-3» was found at a confidence threshold of around $0.9$. At higher threshold values the gradient of the recall decrease starts to increase. The maximum of the $F1$ score is not reached there but is reasonably close and some compromise has to be taken. The curve signatures are found to be similar for the validation and test data set results, however the latter are slightly lower either by chance or the fact that the test data set was not used to find the optimal MASK R-CNN model run. At the model confidence level of $0.9$ $F1$ is close to $0.8$ ($0.85$) when evaluating the test (validation) data set. The appearance of 4 groups in the two plots of Fig. 5 is due to the quadruple variation of the learning rate (Table 1).

Looking deeper into the validation data set consisting of $249$ annotated hailstones, we find a $TP$ number of $237$ and a $FN$ number of $12$ which gives a miss rate or false negative rate ($FNR = FN/(FN + TP)$) of $4.8\,\%$. For the test data set ($215$ hailstones) $TP$ reaches $198$, $FN$ count $17$ which yields to $FNR = 7.9\,\%$. The mean average precision, that is calculated over the whole IoU range, for the $90\,\%$ hail confidence threshold reaches $0.53$ (validation data set), respectively $0.50$ (test data set). Figure 6 gives an advanced view on the number distribution of the IoU of the true positive matches (confidence level $C_i \geq 0.9$), again for both the validation (blue bars) and test (green bars) data sets. A large majority of the hail IoUs lie above $0.7$. For the test data set a bi-modal distribution shape is found with peaks around $0.76$ and $0.86$.

As mentioned earlier, a pre-evaluation of the performance of the model capability to produce a reliable HSD has been done by generating two more reference data sets from the 33 test tile images. This also gives the opportunity for a certain evaluation of the references themselves by analyzing the differences between the independent hail annotations. From the annotated polygons the sizes are derived and together with the model results the comparison between four hail size distributions is shown in Fig. 7. For all shown hail size distributions later, a bin size of $3\,\mathrm{mm}$ is taken. The distributions from the model, expert B and expert C peak in the $6$–$9\,\mathrm{mm}$ major axis hail size bin, where also the median ($9\,\mathrm{mm}$) is found. Against this, the peak and the median ($10.5\,\mathrm{mm}$) slip one bin to the right in case of the distribution based on expert A annotations. Overall the discrepancies are largest for the smallest hail major axis size bin ($3$–$6\,\mathrm{mm}$). This probably indicates that the orthophoto resolution limits the reliable identification of those small hailstones by human vision. The clear visibility of many small hailstones suffers from a reduced lightness due to clearer ice and a translucent background. In the main results Sect. 3 and discussions (Sect. 4) we will elaborate on the lightness issue.

## 3 Results

In this section the most important results are presented by means of the complete time-integrated hail size distribution from the $750\,\mathrm{m}^2$ orthophoto area of the first of five drone-based hail survey flights after the passage of the supercell on June 20, 2021.





The derived distribution is very smooth and provides, for this specific case, a more comprehensive picture of the HSD than
with smaller devices, especially of its right tail (largest hailstones).

## 3.1    Estimation of the HSD

The number distribution (logarithmic view) of the hailstone major-axis lengths is shown in the histogram of Fig. 8. The closest
automatic hail sensor HS2 recorded 9 hailstone impacts with a maximal diameter of $14\,\mathrm{mm}$ in a time span of $3\,\mathrm{min}$. Relatively
seen, much more small hailstones were measured by HS2 (likewise for HS3 and HS4) then by the drone. The up-scaled density
is $45$ hailstones per $\mathrm{m}^2$, compared to $24$ hailstones per $\mathrm{m}^2$ (average) for the orthophoto hail survey area. Possible reasons
include, that drone measurements are affected by complete melting of small hailstones and the overlooking of hailstones due
to too low lightness or hiding in the grass surface. The spatial variability of the hail size distribution can also have an influence,
given that the samples were not taken at the exact same location. Our data illustrates the presence of quite large time span
differences for the impacts on the sensors. HS3 recorded a hail duration of $52.5\,\mathrm{min}$ (probably it was hit by another hail cell
shortly after the passage of the main supercell). The hail duration at HS4 ($13\,\mathrm{min}$) was closest to the one estimated at the soccer
field: $\sim 9.5\,\mathrm{min}$.

The HSD measured by drone-based aerial photogrammetry is shown on Fig. 8. It is based on $18209$ hailstones, which is
substantially larger than any sample measured by the near-by automatic hail sensors (Fig. 2(d)). Thus, it provides a much more
comprehensive picture of the shape and upper tail of the HSD. In the orthophoto area most hail objects ($6663$) were classified
in the $6$–$9\,\mathrm{mm}$ bin. Within the large survey area of about $750\,\mathrm{m}^2$, $45$ hailstones are found to be greater than $30\,\mathrm{mm}$ and the
largest hailstone size reached $39\,\mathrm{mm}$. These largest hail sizes are not captured by the hail sensors, as large hailstones are more
sparsely distributed. In the subsequent Section 3.2 we try to asses the sampling error of hail sensors by showing how many
random virtual placed hail sensors would capture hailstones larger than a certain threshold and also the probability for a no-hit
event is investigated.
We note that the mean lightness value (Fig. 8, orange line) increases with increasing hail size. The lightness value is shown
here as a digital value in the HSL color space. The theoretical maximum is $255$ and the highest value is just below $250$. For the
very small hail the mean lightness shrinks below $180$ and thus becomes gradually similar to the lightness of the background.
Edge detection methods based on the lightness value alone, such as proposed in the work of Soderholm et al. (2020), will have
difficulties in finding the correct hail pixel edges.
The same drone-based HSD as in Fig. 8 is shown again with the probability density in Fig 9(a). There additionally the shape
is approximated by a gamma probability distribution function (PDF). In general, the gamma PDF was also found to be most
suited to hailstone major-axis lengths by other case studies, e.g. Ziegler et al. (1983); Fraile et al. (1992). In our case the gamma
PDF slightly underestimates the probability density of the peak, but still a quite smooth fit is achieved. The median size was
found to be $9\,\mathrm{mm}$ (s. Fig. 9(a)). The probability density of the hail aspect ratios shows, that the majority of hailstones show
equal axis lengths (Fig. 9(b)). $75\,\%$ of the hailstones have aspect ratios higher than $0.75$.



## 3.2 Assessment of sampling error of hail sensors using drone-based data

In the previous section, we showed that drone-based aerial photogrammetry can provide a more comprehensive picture of the tail of the HSD than hail sensors, due to its larger sampling area. We note that the HSD is considered at the scale of a single hail cell. The simulation performed in this section assumes that the full HSD is known and given by the drone-based hail data.

Based on this assumption, we investigate the probability that a randomly placed hail sensor on the orthophoto area is not hit at all or hit by a stone larger than a given size. To do so, we randomly placed 10000 virtual areas (blue circles in the orthophoto of Fig. 3(d)) of the same size as the hail sensor ($0.2\,\mathrm{m}^2$) on a $600\,\mathrm{m}^2$ area from the orthophoto. For each virtual area, the HSD was derived. The individual Kernel density estimates (KDE, gray lines) are plotted Fig. 10(a). The KDE could be obtained from 7817 virtual sensor areas. The remaining 2183 sensors had too few impacts and the KDE could not be estimated. The

distribution from the whole area is shown by the black line, and the respective quantiles (Q25, Q50 and Q75) from all the virtual sensors as red lines.

The data from the large random generated virtual sensor samples reveals that only $0.3\,\%$ (34 out of 10000 virtual sensors) record hits larger than $30\,\mathrm{mm}$, $9.9\,\%$ (988 sensors) record hits larger than $20\,\mathrm{mm}$ and $65.8\,\%$ (6576 sensors) record hits larger than $10\,\mathrm{mm}$. Moreover, the probability of a no-hit for an individual sensor was found to be about $4.7\,\%$. While we found 45

hailstones $> 30\,\mathrm{mm}$, the probability for a sensor to record such a large hailstone is only $0.3\,\%$.

In Fig. 10(b) a distribution calculated from the largest hits on each virtual sensor is shown with markers of certain percentiles. The median value reaches $12\,\mathrm{mm}$ and the $95^{\mathrm{th}}$ percentile is at a major axis length of $24\,\mathrm{mm}$.

Figure 10(c) shows the histogram from the number of hits per virtual sensor areas and compares it with the point measurements of the 4 closest hail sensors. The locations of those sensors in context with the drone observations on the soccer field is

shown on the map in Fig. 2. All those sensors were crossed by the $100\,\%$ POH region ($1 \times 1\,\mathrm{km}$ resolution) of the hailstorm. Regarding the $600\,\mathrm{m}^2$ area the probability for 3 hail impacts on a small sensor area was highest (c. peak of the histogram). Zero hits (e.g. HS1 sensor, cyan line) were more likely than 9 or 10 as recorded by the HS2 sensor (blue line) and HS4 sensor (red line). The HS3 sensor towards the SSW with 33 impacts seems to be an outlier here. Very likely, the HS3 sensor was hit by a second hail cell later, because the time interval between the first and the last impact was $52.5\,\mathrm{min}$, compared to $3\,\mathrm{min}$ at HS2

and $13\,\mathrm{min}$ at HS4. The maximum number of hits from the 10000 random sensor placements is 12. But one must be aware that an unknown number of small hailstones were already completely melted and not available in the data set for this investigation with the consequence of a biased histogram towards fewer hail impact numbers.

## 3.3 Melting on the ground and the impact on the estimation of the HSD

The use of drone aerial photogrammetry for measuring hailstones has a drawback - the drone cannot be flown until the hailstorm

has ended. This delay causes a time gap in measuring the hailstones, allowing them to melt on the ground. In this section, we try to estimate the impact of melting by comparing the data from five successive drone flight missions, where the temporal evolution of the HSD could be monitored. As equally sized areas are important for this investigation, we decided to crop the area of the orthophotos to the marked soccer center circle. This procedure was a compromise because a one to one assignment





**Table 2.** Time slots in UTC, when the aerial pictures of the soccer center circle $(263\,\mathrm{m}^2)$ were captured for the five drone mapping flights. The drone image capture intervals with the camera last between 198 and 200 s. The time differences between the sequence of orthophotos are: 307 s, 273 s, 268 s and 271 s. From the first to the last orthophoto 1119 s (18 min, 39 s) passed. The last column gives the number of hailstones detected within the soccer center circle for each of the five orthophotos.

| Capture series | Start [$UTC$] | Stop [$UTC$] | Capture interval [$s$] | No. hail |
|:---:|:---:|:---:|:---:|:---:|
| 1 | 14:37:59 | 14:41:19 | 200 | 3925 |
| 2 | 14:43:06 | 14:46:25 | 199 | 3077 |
| 3 | 14:47:39 | 14:50:59 | 200 | 2511 |
| 4 | 14:52:07 | 14:55:27 | 200 | 1962 |
| 5 | 14:56:38 | 14:59:56 | 198 | 1411 |

of all individual hailstones between the orthophotos was not feasible in a reliable way due to small misalignments and changes
in the location of the center hail pixel due to the melting and differences in the orthophotos. To illustrate this, we show the shape evolution for the five time stamps (s. Table 2) of two large, prominent hailstones in Fig. 11. Those hailstones shrink from initially 33 mm to 21 mm, respectively 25.5 mm, during the course of 1119 s.

With a radius of 10 yards (9.15 m), the area of the soccer center circle is well defined and reaches $263\,\mathrm{m}^2$. The fact that this area was covered in all 5 drone flights makes it an ideal start point to deeper investigate the melting process. With this we are
able to get a first idea how much the upper tail of the distribution degraded and thus approximate better the ground truth of maximal hail sizes. A first observation tells us, that the number of hailstones is roughly reduced by $64,\%$ in the soccer center circle (s. Table 2) from the first to the fifth hail survey. The evolution of the Kernel density estimation for all five orthophoto soccer center circle cut outs is once shown with normal and logarithmic y-axis in Fig. 12. From the black to the red curve we see how the peaks and tails degrade over the distinct time frames. The peak density decreases from more than 0.038 to 0.017,
while moving slightly leftward towards smaller hail size bins. Tracing certain plateaus in the different colored distributions, melting rates in the order of magnitude of $0.5\,\mathrm{mm\,min^{-1}}$ can be deduced. This is supported by the more accurate results from an individual tracking of 48 hailstones of different initial sizes. The melting rate range, that was observed for the different hail size bins, lied in average between $0.3\text{–}0.5\,\mathrm{mm\,min^{-1}}$. The sample size of 48 hailstones however is much too small to make assumptions on melting speed in relation to the initial hail size.

By assuming a melting rate of $0.5\,\mathrm{mm\,min^{-1}}$ and a temporal delay of 9.5 min (time difference between start of hail and drone image capturing) for i.e. the largest hailstone (39 mm), an initial size of rounded 44 mm could be expected, which is closer to the result of MESHS (63 mm). Most crowdsourced reports in the vicinity of the soccer field indicated sizes of a 5 CHF coin ($\sim 3\,\mathrm{cm}$). Golf-ball sizes ($\sim 5\,\mathrm{cm}$) were also reported a few kilometers to the NE of the soccer field (s. Fig. 2(b)). Immediate on site measurements during the hail event revealed maximal hail diameters between 4 and 5 cm as well.



## 4 Discussion

Hail forms through a combination of dry and wet growth processes, which can lead to varying densities and appearances in the ice. Dry growth results in bubbles and irregularities in the ice that scatter light, while wet growth causes liquid to soak into gaps and form a clearer and higher-density ice. Hailstones can alternate between these growth regimes, leading to alternating layers of cloudy and clear ice (Allen et al., 2020; Kumjian and Lombardo, 2020; Brook et al., 2021). For the detection and size estimation of hail stones in image data, these facts are of relevance.

In a first step, a pure computer vision approach without the use of neural networks was tested to extract the binary hail masks. The approach was based on lightness thresholds, morphological transformations and watershed algorithms (Najman and Schmitt, 1994) for image segmentation within OpenCV (Bradski, 2000). The success and reliability of this approach highly depended on the visual appearance of the hailstones. For larger sizes it worked well, but with the decreasing lightness of the small hailstones (Fig. 8) the method produced very poor results like the CV-based edge detection (s. Sect. 1 and 5). It could be different with the appearance of hail clusters on the ground, where algorithms based on watershed could retrieve more reliable information, but this needs to be tested.

In a second step a deep-learning model (Mask R-CNN) was tested. By now, the training of this model consisted of only one single hail class. Performance wise, it might be worth to check if an inclusion of different hail size classes can improve the hail predictions and mask shapes. For a simple check we propose to simply start with two classes: small and large hail. The exact size threshold for separation needs to be defined, but could lie for instance at 20 mm, where the potential for damage starts to rise. A thorough investigation of hundreds of hyper-parameters can lead to better results, but this is out of the scope of this study. If the future direction is to build a more generalized Mask R-CNN model for hail detection and size estimation, it is a good idea to invest more into the tuning of the training and validation configuration.

Splitting the orthophoto into many smaller image tiles can produce artificially cropped hailstones. To avoid this issue, producing overlapping tiles by the maximal length of the largest observed hailstone are one possibility, as implemented by Soderholm et al. (2020). However, for sporadic large hail coverage (no clustering of many hailstones on the ground), as observed for the 2021-06-20 supercell storm, the expected corrections due to the few cropped hailstones are marginal in comparison to other errors like negative and positive false detections. Also a cropped hailstone binary mask can still lead to the correct major axis length.

The acquired hail images from the drone show large differences in their transparency, making it difficult to apply simpler computer vision techniques for detection and size estimation. Beside the effect of melting after hail has reached the ground that can change the color and transparency in the visible wavelength range, the microphysical growth processes in the storm determine the inner and outer structures of a hailstone. Often, analyzed hailstone slices (Soderholm and Kumjian, 2022) show layered patterns with alternating transparencies (clear versus cloudy ice). During dry growth, where super-cooled liquid water freezes immediately onto the surface of the ice particle, the probability for trapped air bubbles is high (Rasmussen and Heymsfield, 1987; Pflaum and Pruppacher, 1979).





**Table 3.** Measurements of temperature at $2\,\mathrm{m}$ ($T_{2m}$), $5\,\mathrm{cm}$ ($T_{5cm}$), ground-level ($T_{0cm}$) and relative humidity at $2\,\mathrm{m}$ ($RH_{2m}$) from the SwissMetNet (SMN) weather station in Langnau i.E. ($744\,\mathrm{m}$ a.s.l.) and measurements $T_{2m}$ and $RH_{2m}$ from the SMN weather station in Schüpfheim ($744\,\mathrm{m}$ a.s.l.). The temporal period is between $14{:}00\,\mathrm{UTC}$ and $15{:}30\,\mathrm{UTC}$ on 2021-06-20 with a resolution of $10\,\mathrm{min}$.

| Time | 14:00 | 14:10 | 14:20 | 14:30 | 14:40 | 14:50 | 15:00 | 15:10 | 15:20 | 15:30 |
|---|---|---|---|---|---|---|---|---|---|---|
| $T_{2m}$ **(Schüpfheim)** | 23.5 | 23.4 | 23.1 | 21.0 | 19.3 | 20.3 | 19.4 | 18.4 | 18.4 | 18.6 |
| $RH_{2m}$ **(Schüpfheim)** | 66.9 | 67.5 | 69.2 | 77.9 | 86.0 | 82.1 | 87.7 | 89.2 | 90.4 | 90.5 |
| $T_{2m}$ **(Langnau i.E.)** | 23.4 | 22.9 | 18.9 | 18.4 | 18.1 | 18.4 | 18.4 | 18.4 | 18.7 | 19.1 |
| $T_{5cm}$ **(Langnau i.E.)** | 22.0 | 21.5 | 17.9 | 17.4 | 17.4 | 17.2 | 16.9 | 18.0 | 19.8 | 21.6 |
| $T_{0cm}$ **(Langnau i.E.)** | 21.8 | 21.3 | 18.1 | 17.4 | 17.5 | 17.3 | 17.1 | 18.1 | 20.2 | 21.9 |
| $RH_{2m}$ **(Langnau i.E.)** | 71.1 | 75.7 | 93.9 | 96.5 | 99.2 | 93.7 | 90.8 | 94.9 | 96.9 | 92.7 |

A few studies exist, that explore the effect of melting hail in the air from polarimetric radar measurements (Ryzhkov et al., 2013; Kumjian and Ryzhkov, 2008) or models (Fraile et al., 2003). To our knowledge there are no studies, that analyze the melting of a large sample size of real hail on the ground after the passage of a hailstorm. Here we provided a potential method to cover this gap and potentially allow to retrieve the original HSD. However, environmental conditions like ground temperature and occurrence of rain before, during and after the hail event can strongly impact the melting rate and therefore would pose some uncertainty to a reconstructed initial HSD.

Because the melting rate will be dependent on temperature and relative humidity, some measurements of those parameters in the relevant time period are shown in Table 3 for two SMN (SwissMetNet) weather stations (Schüpfheim and Langnau i.E.). Although the distances to the soccer field are $5.7\,\mathrm{km}$ (Schüpfheim) and $20\,\mathrm{km}$ (Langnau i.E.) the geographic locations in the same valley are somewhat comparable. In particular, Langnau i.E. is included here because it is the closest station with measurements of temperature at $5\,\mathrm{cm}$ above grass and at the ground. Unfortunately no in situ meteorological measurements at the hail survey area of the 2021-6-20 event are present. Precipitation measurements are available at a distance of $670\,\mathrm{m}$ to the east. There, an automatic rain gauge (Station: Entlebuch) recorded $9.1\,\mathrm{mm}$ between 14:30 and 14:40 UTC and $0.2\,\mathrm{mm}$ in the subsequent 10 minutes. The very light precipitation continues also in the still relevant 10 minutes time slot from 14:50 until 15:00 UTC (compare times in Table 2) with an accumulation of $0.3\,\mathrm{mm}$. Thus, the time-integrated HSD from the first hail survey mission with the drone was most exposed to rain. Temperatures close the ground have fallen by about $4.5\,^\circ\mathrm{C}$ between 14:00 UTC and 14:30 UTC, after the supercell passed the Langnau i.E. SMN weather station. For future drone-based hail surveys that try to retrieve information about melting, it could be a good idea to place a mobile weather station or some ground temperature sensors at the hail measurement location to record more accurate data.

## 5 Conclusions and outlook

Reliable ground truth data from hail observations are rare and of high value to the hail research community. This paper describes the application of aerial drone-based photogrammetry combined with a state-of-the-art deep-learning object detection





model to retrieve the time-integrated hail size distribution over a large survey area. The ability to analyze a large survey area
allows to capture a much more representative sample of the distribution than with other ground-based measurement techniques,
especially for the upper tail of the size distribution.

During a period in June 2021, when exceptionally strong convective supercell storms occurred, a successful data collection
with the drone took place. On 2021-06-20 drone-based photogrammetric data from a sporadic large hail fall (no clustering)
from a right-moving classical supercell could be collected near Entlebuch (Canton Lucerne, Switzerland). Aerial drone im-
agery of the hail survey area ($750\,\mathrm{m}^2$ on a soccer field) could be captured in five subsequent photogrammetry flight missions
between 14:38 and 15:00 UTC. We presented our approach to retrieve the hail size distribution using a deep-learning instance
segmentation model (Mask R-CNN) under the Detectron2 framework. A broader part of the paper dealt with the validation and
testing of the model predictions.

A short summary of the key results and conclusions of the presented work is listed below:

– A robust retrieval of a HSD based on a population size of $18209$ hail stones on an area of $750\,\mathrm{m}^2$ from a single hail event
with a duration of about $9.5$ minutes on 2021-06-20 was carried out. The median size was $9\,\mathrm{mm}$ and hailstones with
equal axis length (minor/major) dominated.

– The largest hail stone reached $39\,\mathrm{mm}$ and is substantially larger than impacts on the closest hail sensors at distances
between $300\,\mathrm{m}$ and $1470\,\mathrm{m}$ from the soccer field.

– A combination of hail data from different applications (drone, sensor and crowdsourced) that observe the same hail fall
improves the reconstruction of the complete HSD of such an event and also helps to frame the individual limitations.
This will help to better compare the data also to radar-based hail products.

– Investigations with virtual sensors can provide relevant statistical information for various applications, e.g. probability
of miss rates and impacts of certain size on a small sensor measurement area.

– The decay of the HSD caused by melting could be monitored during $18.5\,\mathrm{min}$ by performing additional drone pho-
togrammetry flights. Melting rates in the range $0.3$–$0.5\,\mathrm{mm\,min}^{-1}$ were estimated.

The aerial drone footage of a larger field with hail is not an instantaneous picture of hail fall, but prone to different ages of
hailstones after they fell on the ground. Although the time differences are just within minutes, the hailstones are in different
melting stages and appear differently regarding their outer ice transparency. If the aim is to capture all of those different looking
hailstones for the best possible estimation of the distribution, we find that the two-stage approach from Soderholm et al. (2020)
combining machine learning for detection and computer vision for size retrieval, gives not an optimal result. The edge detection
algorithm on the pixel lightness fails in our case for hailstones where the edge regions show a clear ice structure and the
lightness becomes similar to the background. The same problem will be present for even simpler CV approaches without the
use of neural networks, e.g. blob detection (Lindeberg, 1993). If the background and objects would be homogeneous enough,
identifying the correct blobs and shapes is straight forward.





**Table 4.** Schematic list of some advantages (green) and disadvantages (red) of the two hail observation methods: Drone-based photogrammetry and automatic hail sensor. With the «clustering problem» we refer to the issue, when too many hail stones are close to each other and the separation of the individual hailstones is getting difficult in the image data. On the sensor side, an equivalent problem is a dead time after each hail impact. In a worst case of fast, subsequent or simultaneous hits a separation of those is not possible.

|  | **Drone-based photogrammetry** | **Automatic hail sensor** |
|---|---|---|
| **Sampling error** | low | high |
| **Melting problems** | yes | no |
| **Exact time information** | no | yes |
| **Probability to capture largest hailstones** | high | low |
| **Daylight dependence** | yes | no |
| **Operational application** | difficult | easy |
| **Clustering problems** | high | existing, but low |
| **Size estimation** | direct | indirect |

Drone imagery acquisition could be improved, considering that low light levels can be a main issue whenever the thunderstorm occurrence drags on into the night or the available light is greatly reduced by the presence of the thunderstorm itself. The amount of light present in a scene governs the required exposure time given all other camera settings are constant. The resulting exposure time essentially limits the maximum flight velocity given a sub-pixel motion blur criterion. Adding more light in the form of a drone-mounted flash could allow for an increased flight velocity while conserving the sub-pixel motion blur condition, allowing for more area to be covered in the same time. More light would also improve image quality, as the sensor gain (ISO) and the aperture size could be reduced.

Nowadays, with the high availability and coverage of radar observations and sophisticated hail products, like MESHS, the demand for objective ground truth observations of the hail size is rising to allow verification studies of the radar-based algorithms. Beside crowdsourced observations, traditional hailpads and new automatic sensors, field observations with drones can be a very useful additional information source of hail size data. The possibility to spatially collect hail size data with drone-image photogrammetry over quite large areas gives new insights into the HSD of hailstorms. Here we presented the large discrepancies between MESHS and the hail sensor data and explain that for specific hail events the drone-based hail data can provide additional information and thus complement automatic hail sensor measurements. Both measurement applications have their advantages and drawbacks (s. Table 4). The sensor provides exact time information and is not affected by any melting on the ground, but the limited area covered leads to truncated distributions. The drone-based approach with the large area allows for a more representative sample of the hail size distribution, but is impacted by melting and image quality. Using those measurement sources in combination to observe the same hail fall could improve the reconstruction of the complete HSD of such an event and also help to frame the limitations of both applications. In this sense, it could be a good idea to carry along a few traditional hailpads during storm chasing, which could be deployed right before a hail fall so that those measurements could be compared as well with the drone-based hail data.

To further assess the hail size distribution of different storms, more observational data is crucial. However, as described in the introduction (Sect. 1), the collection of drone-based areal photography is a time consuming and difficult task. Therefore it could be beneficial to set up a database of drone-based maps for hail surveying to further adapt and test the existing algorithms. In addition, with the increasing usage of personal drones equipped with cameras, there could be a public community that has the basic requirements for such observations. It might be useful to provide the requirements about how to collect adequate image data and use it in a crowdsourced approach similar to the existing crowdsourced information retrieval at weather services (e.g. German Weather Service DWD and Federal Office for Meteorology and Climatology MeteoSwiss).

Another point to stimulate in future could be tests with artificial hail objects of defined size on real backgrounds such as short mowed meadows. In this way several setups can be trained, tested and optimized: Save and smooth drone operation in various conditions, flight missions and camera settings and precise comparison of the retrieved HSD to the known ground truth.

*Data availability.* The drone-based hail size data set of the 2021-06-20 event are available on request.

*Author contributions.* ML performed the following roles: conceptualization, methodology, software, validation, hail annotation, formal analysis, visualization, and writing the original draft. KB performed the following roles: conceptualization, methodology, storm chasing, drone operations, review and editing. AH performed the following roles: PI hail sensors, review and editing. JK performed the following roles: hail sensor data preparation, review and editing. SM performed the following roles: conceptualization, methodology, storm chasing, hail annotation, review and editing, and project administration. DW performed the following roles: conceptualization, methodology, hail annotation, review and editing. UG performed the following roles: initiation of hail research projects, acquisition of funding, procurement of equipment, review and editing.

*Competing interests.* All author declare to have no competing interests.

*Acknowledgements.* Hail is a severe threat to the society and on-going research is important to be able to establish risk mitigation measures. In this context, we thank the Swiss Insurance Company La Mobilière for funding the automatic hail sensors network and making the hail sensor data available for research investigations. We want to acknowledge the fruitful scientific exchange with Joshua Soderholm (Australian Bureau of Meteorology) about drone-based hail photogrammetry.



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

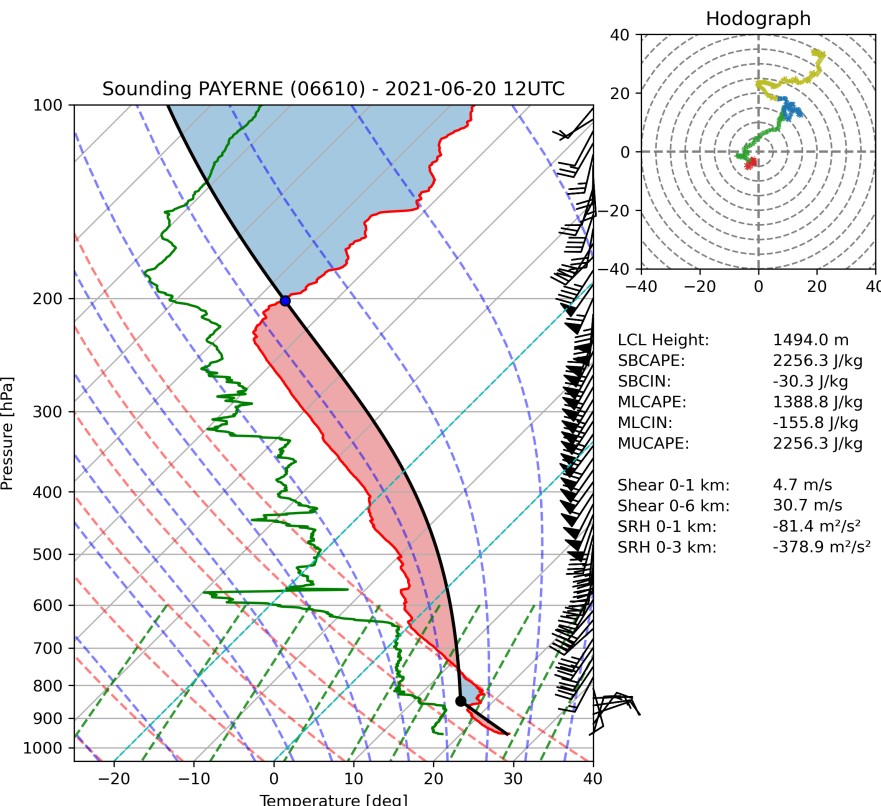

**Figure 1.** Skew-T plot with hodograph analysis from the atmospheric radio sounding at the Payerne station (ID: 06610, 87 km WSW from the soccer field) on 2021-06-20 12 UTC. The temperature and dew point profiles are drawn in red and green. The shaded areas in red and blue mark the CAPE (Convective Available Potential Energy) and CIN (Convective Inhibition). The sounding is characterized by a moist, fairly well-mixed layer, separated from a dry layer above by a capping inversion. Lapse rates above the cap are close to dry adiabatic. In meteorology this kind of sounding is also known as a «loaded gun» sounding.

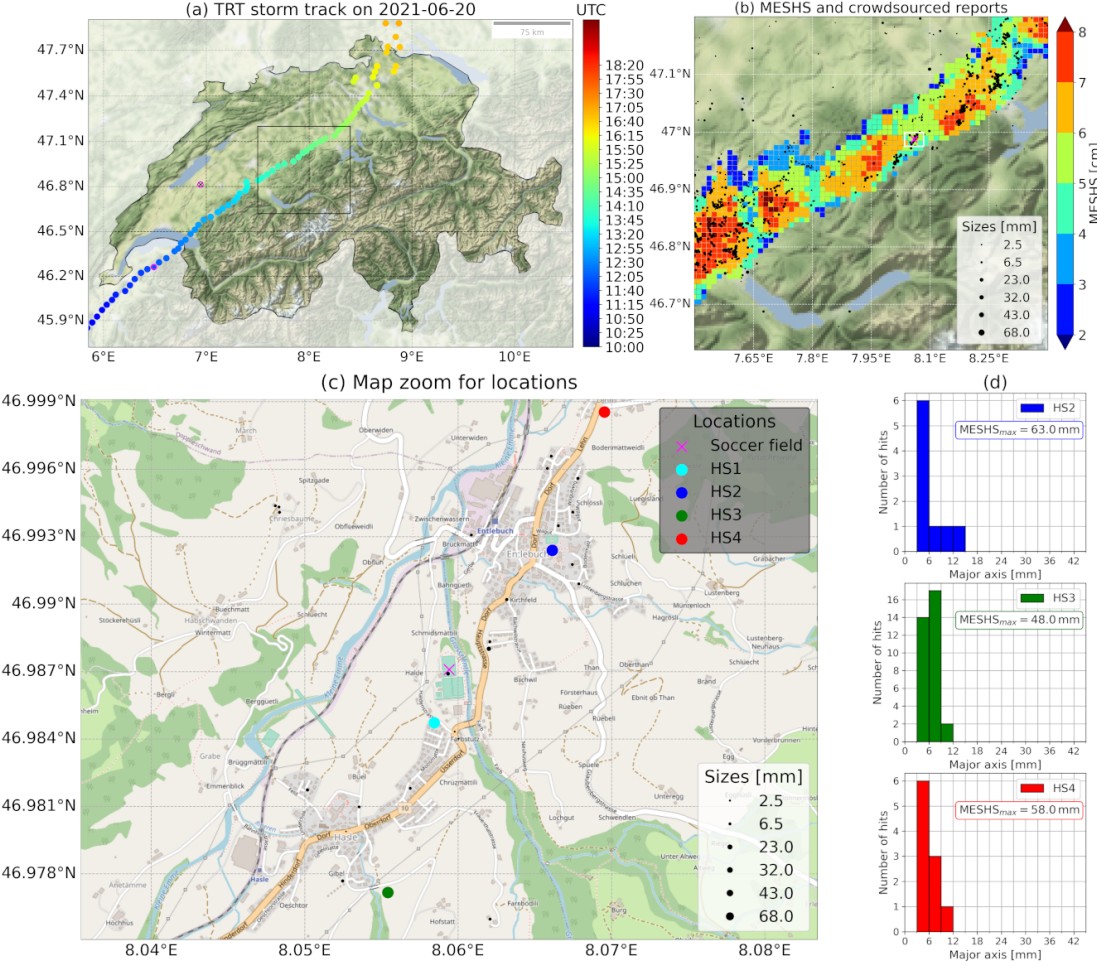

**Figure 2.** Storm track (a) of the 2021-06-20 supercell with colored time information (5 min resolution of the scatter points) and the location of the atmospheric radio sounding (magenta open circle with black cross inside) shown in Fig. 1. The storm location at the sounding time (12 UTC) is marked with the same edge color (magenta). The black rectangle in (a) marks the zoom area for plot (b), where information on radar derived MESHS (Maximum Expected Severe Hail Size) and crowdsourced hail size reports (black and different sized circles for 6 size categories with bin centers at 2.5, 6.5, 23, 32, 43 and 68 mm corresponding to the MeteoSwiss app categories: smaller than coffee bean, coffee bean, 1 CHF coin, 5 CHF coin and tennis ball), are given. The location of the soccer field, where the drone-based hail survey took place, is marked with a magenta cross. The white rectangle around the magenta cross in (b) marks the zoom area for the map view in (c), where the detailed locations of the soccer field (roughly centered to the map view, magenta cross), the 4 nearest automatic hail sensors (HS1, HS2, HS3 and HS4) and the crowdsourced hail size data (black and different sized circles) for this area are shown. The histograms in (d) present the recorded HSDs from the automatic hail sensors together with the daily maximum MESHS value at the sensor locations. The recorded hail duration for the sensors are 3 min (HS2), 52.5 min (HS3) and 13 min (HS4). The HS1 sensor (cyan color) did not record any hailstones, and is thus omitted in plot (d).

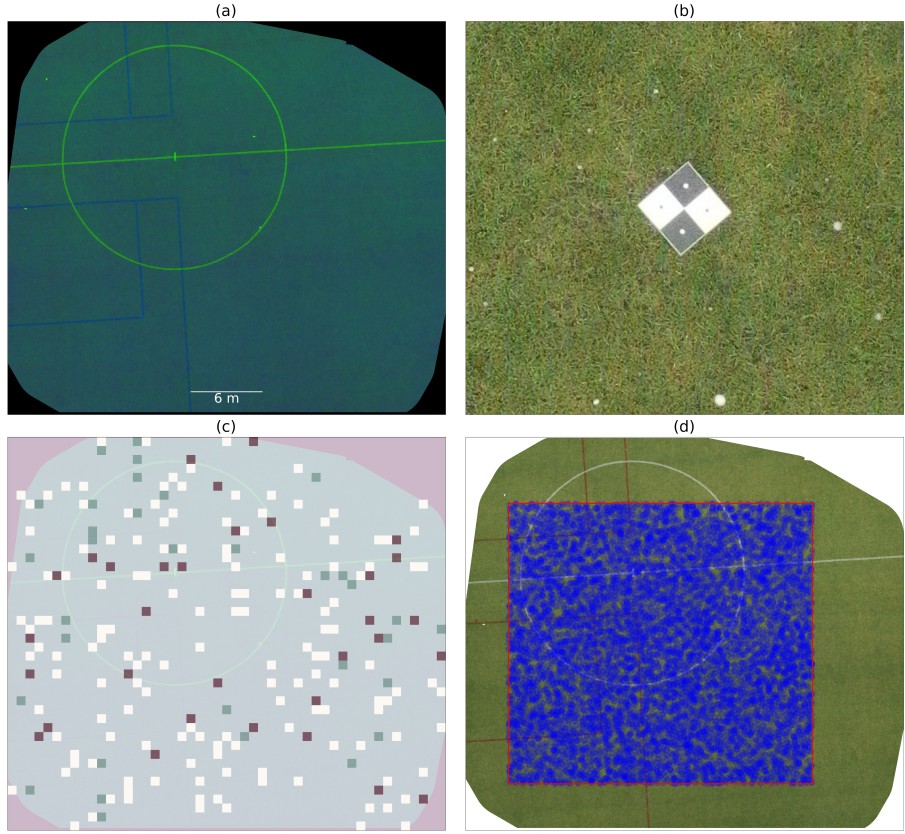

**Figure 3.** In (a), the final orthophoto of the 2021-06-20 hail event is shown in HSL (Hue, Saturation, Lightness) color space. It is produced from 116 individual aerial drone images with the OpenDroneMap (ODM) software package. The radius of the soccer middle circle is $9.15\,\mathrm{m}$. In (b) an image zoom from the orthophoto with actual scale of $1\,\mathrm{m}$ (width) and $0.9\,\mathrm{m}$ (height) illustrates the hail appearance on the soccer field in conjunction with one of the reference objects (black and white circles: $10\,\mathrm{mm}$ diameters; black and white squares: $75\,\mathrm{mm}$ side lengths) to verify the ground sampling distance (GSD). In (c), the random selected distribution of training (whitish), validation (greenish) and test (reddish) image tiles ($75\,\mathrm{cm}$ edge length) are displayed within the orthophoto. In (d), the same orthophoto in RGB (Red, Green and Blue) color space is shown and over-plotted by a $600\,\mathrm{m}^2$ area (red rectangle), where 10000 circles of $0.2\,\mathrm{m}^2$ (virtual hail sensors, blue shaded) are randomly placed for statistical assessments.

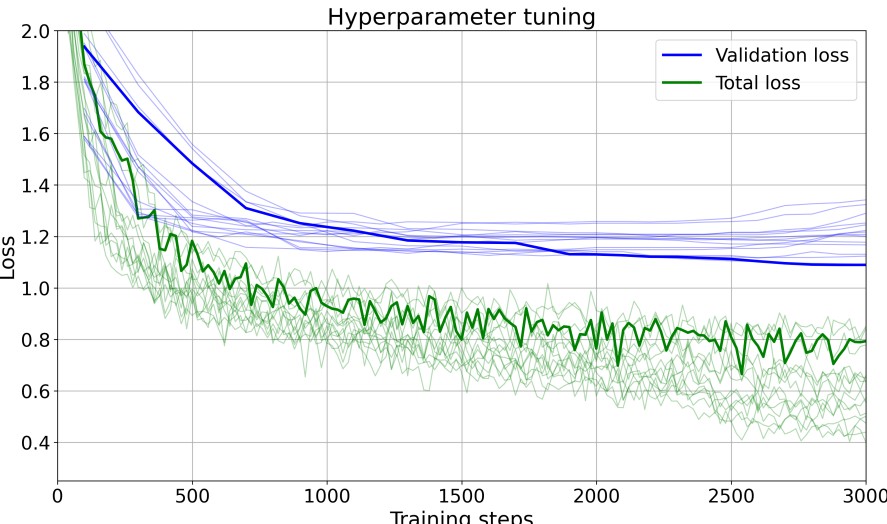

**Figure 4.** Spaghetti plots of the evolution of validation loss and total loss along the training iteration steps for the 16 deep-learning model runs with different combinations of hyper-parameters shown in Table 1. The thick lines depict the training «run-3», used for prediction of hail pixels.

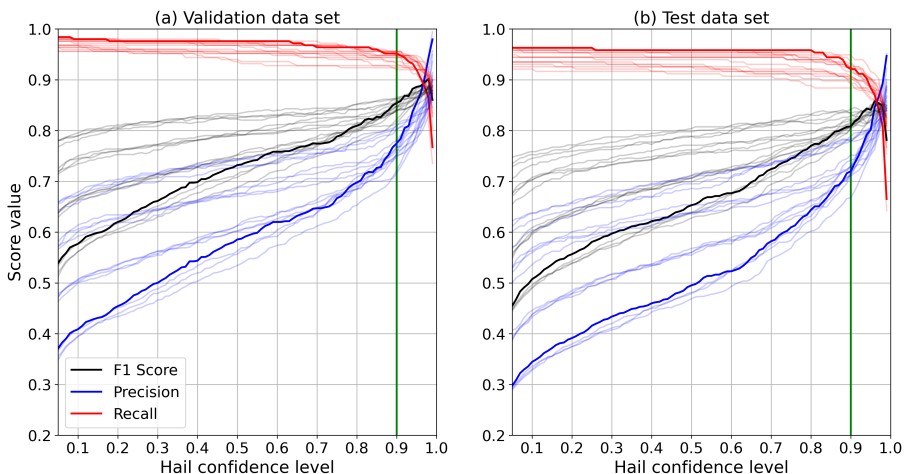

**Figure 5.** Spaghetti plots of precision (blue), recall (red) and $F1$ scores (black) against the hail confidence level for all 16 deep-learning model runs applied to the validation data (a) and test data (b). The thick lines depict the training «run-3», used for prediction of hail pixels. The green vertical line marks the $90\,\%$ confidence value, that has been chosen as the lower limit for the object classification.



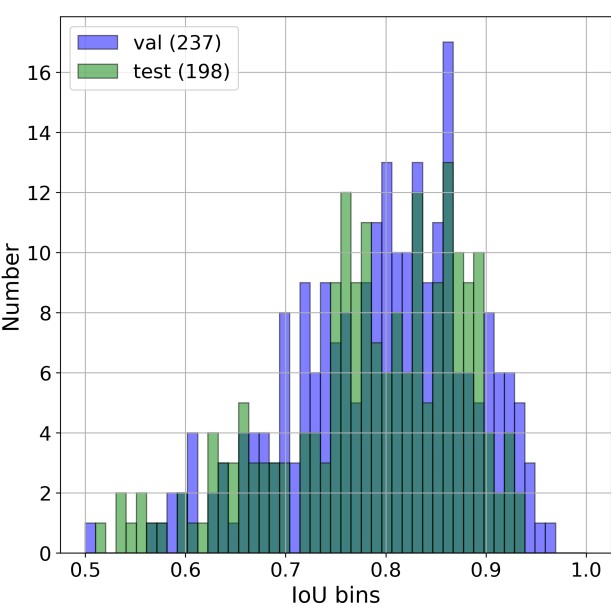

**Figure 6.** Histograms of IoU (Intersection over Union) ratios between model «run-3» prediction masks (confidence $C_i \geq 0.9$) and the validation data set (blue), respectively the test data set (green). The histogram area of the overlap between green and blue bars appears in dark green color. Only true positive ($TP$) matches, defined as IoU $> 0.5$, are shown. In the validation (test) data set 237 (198) hailstones are classified as $TP$.

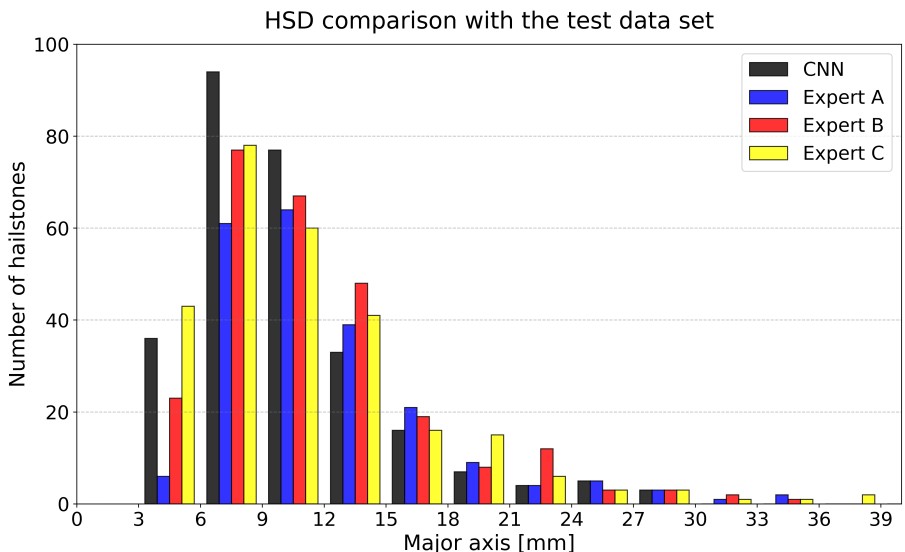

**Figure 7.** Comparison of four hail size distributions (HSDs) from the test data set derived from manual annotations by three experts (A: blue, B: red, C: yellow) and the prediction of the Mask R-CNN model (black). The total number of identified hailstones by the experts are 215 (A), 263 (B) and 269 (C). The CNN (Convolutional Neural Network) predicted 275 hail segmentation masks.

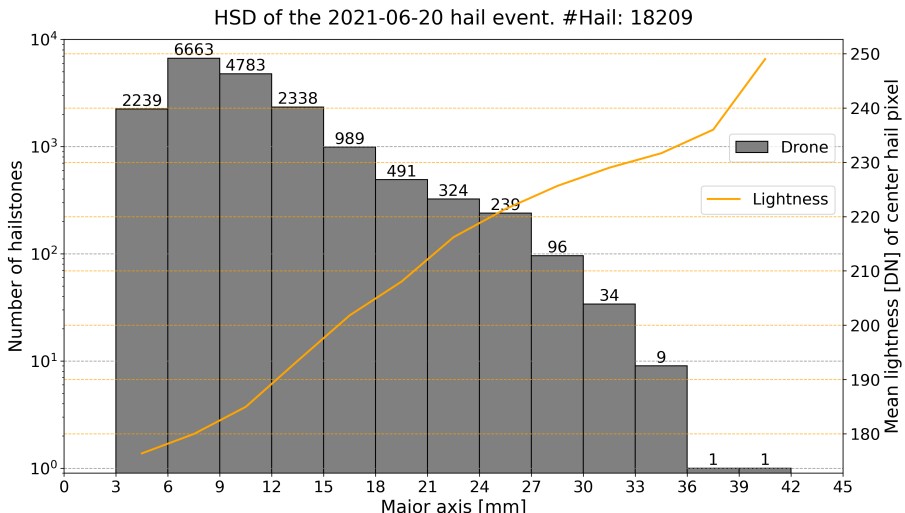

**Figure 8.** Logarithmic view of the time integrated hail size distribution of the 2021-06-20 event captured by the drone between 14:37:28 and 14:41:19 UTC. The total number of detected hailstones per each bin is shown with the number above each bar. All together 18209 hailstones were identified. The orange line represents the mean lightness value as digital number (DN) of all derived center hail pixels in the HSL (Hue, Saturation, Lightness) color space for each hail size bin.



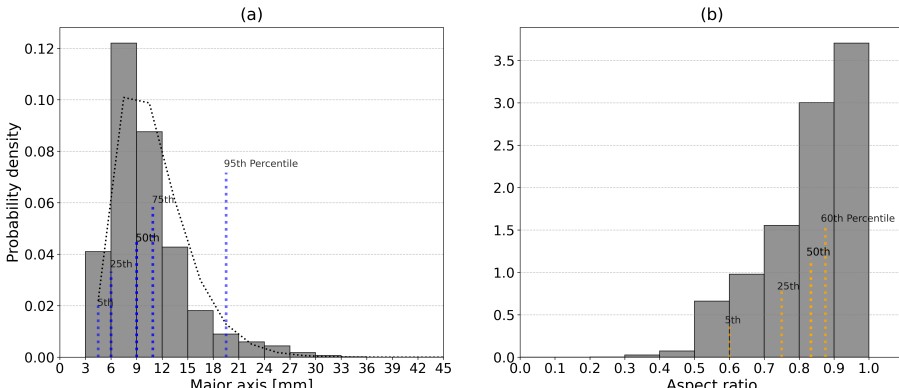

**Figure 9.** Probability density distributions of the hail major axis (a) and the aspect ratio (b) between minor and major axis length. The vertical blue and orange dashed lines indicate the position of the particular percentiles regarding the two X axes. The HSD in plot (a) is additionally fitted against a gamma distribution (black dotted line).

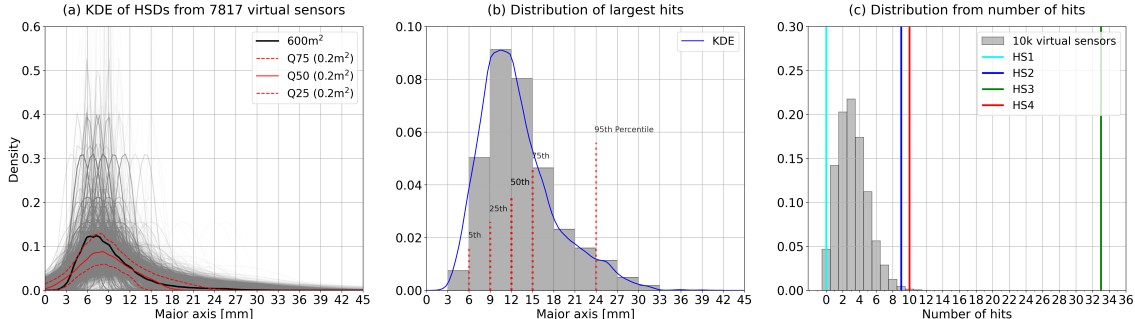

**Figure 10.** Kernel density estimation (KDE) of HSDs (Hail Size Distributions) from virtually and random placed hail sensors (a) on an area of $600\,\mathrm{m}^2$ (red rectangle in Fig. 3(d)). From the 10000 virtual HSDs 7817 can be represented by a KDE (gray curves), whereas the others do not have enough impacts. The quantiles of the sorted HSDs are shown as dashed (Q25, Q75) and solid (Q50) red curves. For comparison, the KDE as derived from the whole $600\,\mathrm{m}^2$ area is overplotted in black. In the center (b), the KDE distribution for the aggregation of the largest hailstone impact on each virtual sensor is shown. Additionally, various percentile markers are drawn on top of the (b) plot in dashed red vertical lines. On the right side (c) the probability density for the total hits on each virtual sensor is shown as gray histogram, together with the registered number of impacts of the four closest automatic hail sensors HS1 (cyan line), HS2 (blue line), HS3 (green line) and HS4 (red line).



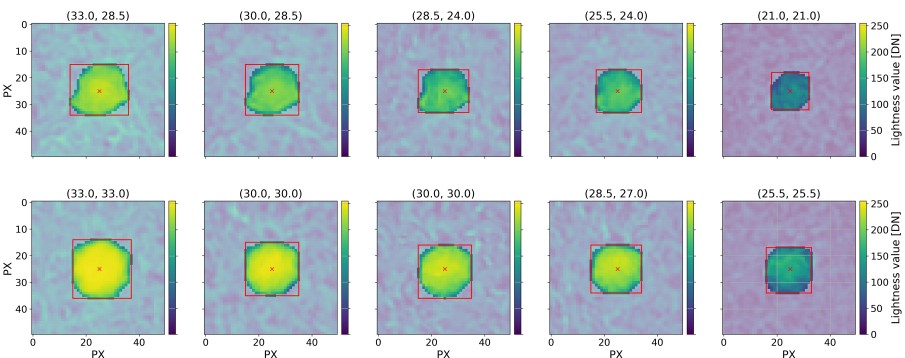

**Figure 11.** Two examples of hailstone size and mask shape development during the captured melting process on the ground. From left to right the sequential lightness images of two hailstones (row 1 and 2) extracted from the five orthophotos (soccer center circle) are shown. In the images the Mask R-CNN segmentation masks are emphasized together with with the major and minor axis lengths indicated by the minimal bounding boxes. The actual sizes (width and height in mm) are given in the titles. During the $1119\,\text{s}$ these hailstone shrink about 12 and $7.5\,\text{mm}$ in their major axis length.





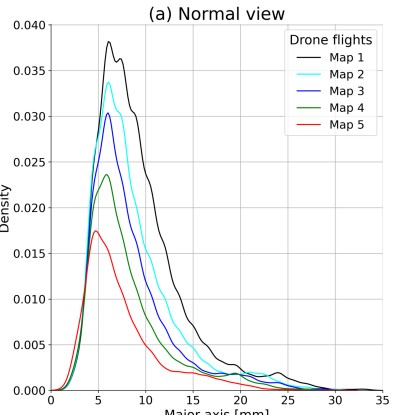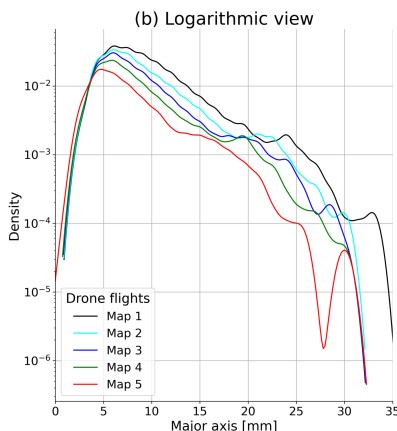

**Figure 12.** Kernel density estimation (KDE) with linear (a) and logarithmic (b) y-axis of the degrading hail size distributions due to melting processes on the ground. The orthophoto area for the melting analysis is restricted to the soccer center circle to ensure a correct comparison between the different orthophotos (Map 1–5). In total, five drone-based hail photogrammetry surveys were carried out to secure the temporal data analysis. All the relevant time frames are listed in Table 2.