# Peer review of "Drone-based photogrammetry combined with deep-learning to estimate hail size distributions and melting of hail on the ground"

_Atmospheric Measurement Techniques, 2023_

## Author Comment (AC1)

**Response to comment on amt-2023-89 by Anonymous Referee #2**

This study present a significant advancement on the initial drone-based technique developed by Soderholm (2020), in particular the derivation on melting rates to estimate the original HSD and simulation of hail pad measurements. This is really great work! The authors also improve the underlying methodology and explore the parameter tuning in much more details.

We very much appreciate the positive feedback by Referee #2 and all the raised comments, that help to improve the manuscript. All Referee comments are shown in black, the authors responses in green text color.

It was unfortunate to see the very serious findings of plagiarism from R1. Further, I believe the manuscript needs to be reviewed by a scientific English language editor first before resubmission. The corrections to make to the language were too numerous that I did not document them and instead tried focused on the science content. The text also uses many superfluous words and sentences with repetition that could be removed, significantly reducing the length. I'd encourage the authors to take these actions on board and prepare a resubmission.

Indeed, there are several instance in the manuscript where the text directly referred to an available source without correct citation. We admit that this should be clearly avoided in a scientific publication and want to thank the reviewers for pointing this out. Please refer to the answer to comments by Referee #1 on this issue.

Regarding the raised language and readability issue, the authors of the manuscript tried to improve the whole manuscript text. The mark-up version looks quite messy, but we note that content-wise no critical things changed. We further hope, that you understand, that we cannot go into detail and list all language-based changes, rearrangements and reductions of the text here in this reply.

We also want to note, that at a later stage (if the manuscript gets there), AMT provides the typesetting and English language copy-editing as a service, which should give the final linguistic polish of the manuscript.

Comments/corrections:

Line 24: "asses"

This typo has been corrected.

Line 38-39: Crowd sourced data might not even provide the largest diameter, just the size at some unknown percentile

This is certainly true. Due to rearrangements the updated version about this issue is now located in the second paragraph of the introduction and reads:

"…In addition, hail sensors cannot capture the entire hail size distribution (HSD) due to their small observational area of 0.2 m (Kopp et al., 2023). Similarly, crowdsourced hail reports use predefined

categories (no hail, < 10mm, 10mm, 20mm, 30mm, 50mm and > 70mm) for estimating the hail size, corresponding to an unknown percentile of the actual HSD."

Line 59: The short sentence about chasing needs some more context - who and why?

Due to thorough language changes, text reformulations and reductions, this sentence disappeared. We also do not use anymore the word "chasing" or "chase" in the text.  It only remains in the title of Section 2.1.

Line 59: How was the supercell track generated in figure 2a?

The cell track of Figure 2a is based on the TRT (Thunderstorm Radar Tracking) method by Hering et al. (2004). The method is also described in Feldmann et al., 2023. We added the corresponding citations here (lines 52-53):

"The track of the supercell shown in Fig. 2(a) and was generated based on the TRT (Thunderstorm Radar Tracking) algorithm (Feldmann et al., 2023; Hering et al., 2004)."

Line 62: Please add in the values for storm motion to provide some context for comparing against other events.

Thanks for the comment, this is indeed useful information to add. We added now the values for the Bunkers Right storm motion vector and the SFC-6km mean storm relative winds:

"From the hodograph shown in Fig. 1 a storm motion vector of 234° at 13 m s$^{-1}$ (according to Bunkers et al. (2000)) and mean storm relative winds (0–6km) of 71° at 9 m s$^{-1}$ can be derived."

Paragraph ending line 75: I think some work is needed to improve the flow between this paragraph and the next perhaps by moving the limitations of hail impact sensors here as motivation for aerial surveys

We totally agree, that here the readability was not very smooth between these paragraphs. The structure in the introduction also changed a lot. Now the motivation for the new technique introduced by Soderholm et al. 2020, follows the paragraph about the limitations of hail impact sensors and crowdsourced reports. We continue then in line 40 with:

"In order to overcome some of the limitations of automatic hail sensors and crowdsourced reports for estimating the HSD, a new technique, called HailPixel, has been introduced by Soderholm et al. , 2020. They propose …"

Sentence starting line 78: Some duplication in this sentence around "melting", which is mentioned twice

Here we now deleted the term "… to prevent further melting", as it is stated already in earlier within this sentence.

Line 81: Please keep units consistent, either mm or cm

Thanks you for this comment. We now always use mm for the hail dimensions/sizes, also for the mentioned MESHS and crowdsourced values.

Line 91: The use of "chain" could be improved with "methodology chain"

Instead of "end-to-end chain" we now write in line78: "In Sect. 2 the methodology is presented, …".

Start of section 2: I find it's often clearer for the reader to be more direct, e.g., "Here we first go into the challenging part" to "First we discuss the"

With respect to a Referee #1 comment, the first 2 sentences at the start of Section 2 were removed. Further, the start of Section 2 now reads different due to rearrangements and language editing of the text.

Lines 119-128: I don't think this paragraph is necessary to support this paper. The methodology of storm chasing is quite specific to the region and the individual.

We agree, that not all information given in this paragraph is needed. Therefore it has been shortened and revised, also with respect to a Referee #1 comment. However, it could be worth sharing of some of the methodological aspects.

Lines 142-147: I feel this is more a reflection on the storm chasing approach that is specific to the authors experiences. It's not necessary to support this paper.

Thanks, and yes we agree, that this paragraph can be removed. It is removed now.

Line 150: Earlier it was started that a 50 MP full frame camera was used. Also MP needs to be expanded.

We checked the specifications and instances in the text. The camera has 45 megapixel and this has been corrected in the abstract. Instead of MP we write now always megapixel.

Line 180: That's an extremely high ISO! Were there any issues with noise or over exposure?

A high ISO value was needed to keep the shutter speed fast and reduce motion blur also in darker scenes during/after the hail storm passed. To keep the flow smooth for the different flights with less camera setting changes, we directly started with such a high ISO. Of course the noise is high but still acceptable for the purpose. For sure many cloudy (whitish) hailstones were overexposed with these settings, but it does not impact the detection of the hail.

Line 183: I'm not sure what the convention is for AMT with notation for number (either comma for full stop). Might be worth checking. Also this needs to be made consistent throughout the text as there's many numbers written within any separator.

During typesetting this will be corrected by the journal to their standards. We had a look through the manuscript so that we are at least consistent in the manuscript for now.

Line 203: ResNet should be expanded in the previous paragraph where it is first introduced

Thanks for the hint, we now expand ResNet (Residual Neural Network) in the previous paragraph: "…uses a Residual Neural Network (ResNet) detection engine described in He et al. (2016)."

Line 221: The sentence starting "The idea behind…" can be merged with the next sentence to make the text less verbose

Some modifications were done to this paragraph. Now this part of the text reads:

"We use 10% randomly selected tiles as reference data (216 tiles). This reference data is further divided into 70% for training (150 tiles) and 15% for the validation (33 tiles) and test data (33 tiles) respectively."

Line 231: "tow"

Corrected to "two".

Line 244: It might be worth explaining to the reader how the term epoch is used for deep learning

For better readability and understanding quite some adjustments have been performed in Section 2.3.2. The epoch time is now better explained within the context:

"Using two images per batch on 1 GPU, a total of 75 batches are needed which represents one epoch time, i.e. to iterate through all available image tiles."

Lines 259-260: I feel this statement about the methodology is already covered in 2.3.1 - "Because we want the test data set to be locked down until we are confident enough about our trained model, we do another division and split a validation set out of the train set. In this scenario we end up with three data sets." Further, "train" should be "training"

The mentioned statement is somehow obsolete and anyway removed/changed now. Where applicable, "train" is substituted with "training" in the right context.

Line 268-269: How extensive was this manual QC to remove non-hail objects? It might have been worthwhile including some tiles with these uncommon non-hail objects in the training.

We agree, that for future model trainings this could be an opportunity to better fine tune the model to distinguish hail from other objects in the image. In our case here it was not very extensive to remove those objects, as there were less than 10. Probably also too few to reliably train on these objects.

Paragraph 270: I think this could be shortened significantly by outlining the parameter space of F1 with a reference.

We thank the referee for his advice to reduce the length of the paragraph following line 207ff.

After major rearrangements and text editing the new paragraph with added references reads now:

"In machine learning, precision and recall (Eq. (1) and Eq. (2)) are commonly used (Powers, 2020). Precision depicts the number of true positive results divided by the total number of positive results. Recall refers to all true positive results divided by the number of all samples that should have been classified (i.e. as visually identified by the experts in the test data set in our case). Precision and recall can be combined in the F1 score in Eq. (3) (Van Rijsbergen, 1979; Goutte and Gaussier, 2005). The F1 score results in values from 0 to 1 where 0 indicates extremely poor performance and 1 refers to a perfect performance of the model."

line 287: "quadruple variation of the learning rate" could be improved with "four different learning rate values tested"

Yes, your suggested reformulation sounds better and is included now:

"The appearance of four groups in the two plots of Fig. 7 is due to the four different learning rate values tested (Table 1)."

Lines 310: "Right tail" should be "Upper tail". I'd also be more specific than saying "smaller devices"

We are now more specific and mention the hail sensor data instead of small devices and use the "upper tail" expression instead of "right tail" throughout the manuscript.

Lines 312: "logarithmic view" is not needed in the main text, this belongs in the figure caption.

Correct not really needed here and thus removed.

Line 313: check use of 'maximal' I think maximum is more suitable.

Thanks for this hint. We checked the whole manuscript for "maximal" and replaced it with "maximum".

line 343: I'm unclear what the author is trying to assert with "We note that the HSD is considered at the scale of a single hail cell."

We agree with the referee that this sentence is confusing. We have rewritten the introduction of this Section 3.2:

"In this section, we determine the probability of impacts of a given hail diameter on randomly placed hail sensors. 10000 virtual hail sensors with a size of 0.2m2 were distributed across an area of 600m2 within the orthophoto (blue circles in Fig. 5(d)). For each virtual sensor, the HSD was derived and the individual Kernel density estimates (KDE, gray lines) are shown in Fig. 12(a). The KDE was obtained from 7817 virtual sensor areas. The remaining 2183 sensors did not have enough virtual impacts to estimate the KDE. The distribution from the entire 600 $m^2$ area is shown in black, and the respective quantiles (Q25, Q50 and Q75) from all the virtual sensors in blue, red and green."

line 354-355: repeated from the start of the paragraph. This is some really nice work too!

Thank you very much! This part was already adapted with respect to a Referee #1 comment.

line 363-364: Can you use the time series information from these disdrometers to separate the two hail events for HS3?

After looking at the time series of impacts for HS3 below (no disdrometer data), the single impact near 15:10 was discarded as it happened 40 minutes after the event and could result from a non-hail object.

[Figure]

All others impacts were considered to be part of the same event as they were separated by less than 5 minutes each, strongly indicating that they belonged to the same cell (Ref: How observations from automatic hail sensors in Switzerland shed light on local hailfall duration and compare with hailpad measurements, Kopp et al. 2023).

lines 376: please avoid repeating information from the caption "Those hailstones shrink from initially 33 mm to 21 mm, respectively 25.5 mm, during the course of 1119 s."

We keep the information in the caption and removed the sentence in line 276. The whole text are here is also strongly rearranged and reformulated.

lines 385: I feel a more effective plot for the analysis of melting rates would be to use initial size bins, fit a linear fit to each size bin and plot the slope. This would directly show the melting rate for different sizes.

Your explanation for a more effective plot of the melting rate is not totally clear to us and we encourage you to explain it a bit more detailed. At least we tried to go into this direction somehow and therefore want to present a plot here in our answer, but we are not sure if this is what you really meant.

In this plot we show hail counts versus time since first capture for a series of bin sizes. The colored scatter points mark the 5 consecutive flights while the colored lines go with the different bins. Of course one could intuitively think that when the slope of these lines is higher, the higher is also the melting rate within this bin. But this is not necessarily true, because stones change bins and can even over jump a bin between 2 consecutive flights. It is quite obvious in the first bin (3-6mm), which gets steadily filled up by stones from the higher bins and we just do not know exactly from which bin these stones come. Most of them likely from the bin above.

[Figure]

line 388: This analysis of 48 hailstones doesn't seem necessary as you can't confirm a robust result and isn't completely described (where is the hail from, what sizes, etc).

We agree, that this analysis is not profound enough yet to be included in this paper and we thus remove the sentences in lines 387-389. The listed results in the conclusion were adapted accordingly:

" - The evolution of the HSD caused by melting could be monitored during 18.65 min by analysing data from multiple drone flights. A melting rate in the order of 0.5 mm min$^{-1}$ could be estimated."

Section 3.2: I'm curious how this experiment would go considering only severe hail sizes (e.g., above 20 mm). But there might not be enough information from the hail disdrometers for a comparison.

We do not use disdrometer data, and assume you mean the hail sensor data. As the HSDs in Fig. 2(d) (new Fig. 4) show, there are no hail diameters recorded on the sensors above 20 mm and as you say we thus cannot compare such results.

Section 4 paragraph 1: I don't think this opening paragraph is needed as this information is discussed again later. Further, it doesn't flow well into the second paragraph.

line 395-400: I would clarify that dry growth produces high densities of microscope air bubbles. Wet growth definitely soaks, but it also accretes on top of existing outer later too.

For now, we decided to keep the paragraph but we reformulated it, also to clarify better the raised points. Now it reads:

"A major challenge for drone-based photogrammetry of hail is related to the appearance of the hail within an orthophoto. The hail stones need to show distinct differences from the background. This is not always the case as hail is formed by a combination of dry and wet growth processes, which can lead to varying densities and appearances in the ice. Dry growth produces high densities of microscopic air bubbles that scatter light, while wet growth causes liquid to soak into gaps and accretes on top of existing outer ice to form clearer ice."

line 401: "pure" isn't needed here. Also "In a first step" should be "In a first attempt". Please also update "Second step" in line 408.

We simplified these sentences (line 345 and 352) to:

"First, a simple computer vision approach (without neural networks) was tested to extract the segmentation hail masks. … Second, a deep-learning model (Mask R-CNN) was tested. "

Line 419: I'm unclear how "Also a cropped hailstone binary mask can still lead to the correct major axis length." I would argue that it would lead to a negative bias.

Our wording here was a bit misleading. Of course you are right, in any way this potential cropping reduces either the major or intermediate axis and generates a potential (if the model classifies it as hail) second hailstone. This sentence is removed and we reformulated (starting line 358) it to:

"However in our case large hail was sparse and, as the image tiles cover large areas (500×500 pixels), it is safe to assume that the number of truncated hailstones is very low. Other sources of errors such as false positive detections or missed hailstones likely play a more important role."

Line 422: I wonder if soaking during melting is the main driver of changes in brightness

Regarding one of your comments below, this paragraph was removed.

Line 428: Ryzhkov et al. 2013 uses simulations of melting hail to estimate changes in polarimetric radar information (which is later used to develop a retrieval). So this isn't a radar study of melting hail.

Sorry for the mistake here. We correct and write now:

"The effect of melting hail in the air was studied by Kumjian and Ryzhkov (2008) using polarimetric radar measurements and numerical model investigations were performed by Fraile et al. (2003)."

Paragraph starting on 421: This paragraph feels incomplete. I'd suggest removing it if the authors can't link this into the results.

For now, we follow the suggestion and removed this paragraph.

Line 473: "different ages" should be "the duration".

Due to extensive language changes and text reduction the expression "different ages" disappeared.

Table 4: The information about the different comparison points should be in the text, not the caption.

For sure it is better to move this information into the text part. In the revised manuscript you find it in line 418ff:

"The comparison of drone-based photogrammetry with automatic hail sensors allowed to highlight the advantages and limitations of both approaches in measuring hail (see a summary in 4). We here want to highlight that the clustering problem refers to many hailstones that aggregate on the ground next to each other. This predominantly occurs during hail events with dominating small hail and intense precipitation. The resulting hail clusters pose a problem to the algorithm to differentiate between individual hailstones. An equivalent problem within the automatic hail sensor data is related to the dead time after each hail impact. The dead time is necessary to avoid any interference with subsequent impacts and to perform the retrieval of the data (Kopp et al., 2023). Furthermore, by combining data from both approaches strongly improves the reconstruction of the complete HSD and could further extend our understanding of hailstorms."

To be compliant with AMT, the red and green color in Table 4 was changed to black. The advantages are now in bold font and the disadvantages in normal font style.

Also the red color in Table 1 was changed to black and bold font style.

Figure 1: Please reduce the number of wind barbs so it's readable! Can you please also annotate the hodograph with the levels or indicate what they are in the figure caption.

As you suggested, we reduced the number of wind barbs in the Skew-T plot. We changed the color for the layers in the hodograph (0, 1, 3, 5, 10 km) and they are defined now in the figure caption. Further the label on the hodograph axis was added and we changed from m/s to kt units. All unit styles are now conform to AMT (no "/" anymore). We removed the interpretation of the sounding from the caption and added the MetPy software reference. Please note there is a slight change in the absolute values for e.g. CAPE, CIN, SRH likely due to the use of a newer version of MetPy.

Figure 2: Please check all the text and annotations in this figure can be read at 100% zoom when rendered. The font size is also not consistent across the subplots. The white box and magenta cross in (b) are not visible at 100% zoom and I also can't find HS1 in subplot (c). Finally, I'd suggest not repeating the same information in both the caption and the main text; for example "corresponding to the MeteoSwiss app categories: smaller than coffee bean, coffee bean, 1 CHF coin, 5 CHF coin and tennis ball), are given." is repeated in both.

As the figure was made out of 4 separate plots and rearranged afterwards it was difficult to get the font sizes consistent. We suggest to split the Figure in 3 separate Figures and keeping only (a) and (b) together with consistent font sizes.

In Figure 2(a) we removed the magenta circle from the storm track as it was hard to see and adjust the caption. The given time information of the cell should be enough to follow the story. Regarding the caption information for Figure 2(b) we removed: ")", as this information is already given in the text part.

HS1 sensor is the closest to the soccer field towards the south and drawn as cyan circle. We enlarged the sizes of all markers a bit (by 20%) and changed from magenta to black cross for the soccer field in old Figure 2(c), which should be better visible. In Figure 2(b) we switched from white to black rectangle for the map zoom area of new Figure 3 (old Fig. 2(c)).

The new Figure 4 shows the HSDs from the 3 hail sensor that recorded hail. Please note, that for HS3 the distribution changed a little bit because we now cut the last impact (likely from a non-hail object). The recording time is now only 16 minutes, rather than 52 minutes, which is also more consistent to the other recording times for the studied hail cell.

Figure 3: Please just describe colors as their proper names (e.g., dark red, light grey and green). Also, there overlap in (d) is significant and I can't really get much from it. Can you just show the center locations perhaps?

Color names were changed from "whitish" to "light grey", "greenish" to "green" and "reddish" to "dark red".

With the added alpha transparency in subplot (d) one at least sea where there is a lot of overlap present as the blue gets darker. From our point of view it makes more sense to show the actual size of the virtual sensor areas with respect to the soccer field area than just the center points of the sensors. However, we also show here the plot version with the smaller center points of the

locations, for further discussion. We are up to add/exchange it at a later stage, if you still think it is worthwhile.

[Figure]

Figure 4: Is this really a spaghetti plot? I would describe this as line plots.

We changed the nomenclature from "spaghetti" to "line".

Figure 8: HSD should be expanded in the title. and # replaced with "hail count"

In Figure 8, HSD is written as "Hail size distribution" and "#Hail" is replaced by "Hail count".

We also changed the title in Figure 7 accordingly to "Comparison of hail size distributions with the test data set" and avoid the acronym HSD.

Figure 9: Font sizes is not consistent for percentiles and I think the colors change? I could extend these lines such that the text sits on top of the highest bars (with no overlap).

The alpha value of the color changed with the percentiles. Now we revised the figure in the way, that we have consistent font sizes for the text which belongs to the percentiles. The text is simplified and made consistent to Figure 10 by the use of e.g. "Q5" instead of "5th percentile". The line color does not change anymore with the quantiles and the height of the lines are now filling the whole plot. Because there can be 2 quantiles on one bar, we should not put the text on top of the bars. The line color for the quantiles in subplot (b) was changed from orange to blue and is now consistent to (a).

In the caption we write: "The vertical blue dashed lines indicate the position of the particular quantiles with respect to the major axis (Q5, Q25, Q50, Q75, Q95) and projected aspect ratio (Q5, Q25, Q50, Q60)."

Figure 10: (a) Q25 and Q75 lines should ideally be different colors. Issue with percentile font sizes again. X-axis labels on (c) are also a bit too close with that font size. Caption: "from virtually and random placed hail sensors" reads better as "from simulated hail sensors at random locations"

Yes, it is a good idea to change colors for Q25 and Q75 in (a). Now they are shown by blue and green dashed lines. The font size for the axis labels has been reduced and now the labels on subplot (c) are better separated. Like for Fig. 9 we now use the quantile nomenclature "Q" and not "percentile" anymore and adjusted the font sizes and the Q-lines (red) fill up the whole y-range of the axis. Please note, the total impacts on HS3 are now 32 (33 before) - see comment on line 363-364.

In the caption we use the terminology of quantiles and we substitute "…from virtually and random placed hail sensors…" with "…from simulated hail sensors at random locations…".

Figure 11: Can you please add the time since first capture above the columns of images? This will be useful to info the reader about the duration since first capture. The final sentence of the caption could be improved with "During the 1119 s these hailstones shrink about 12 mm (upper row) and 7.5 mm (lower row) in their major axis length."

To improve the Figure 11, we included the mm units for the major/minor sizes and we also added the time variable since first capture $t_c$ [s] in the title of each hailstone subplot.

As suggested, the final sentence has been changed to "During the 1119 s these hailstones shrink about 12 mm (upper row) and 7.5 mm (lower row) in their major axis length.".

Figure 12: I don't think the log view adds much value to this analysis because the sample size in the upper tail is so small. The main message is carried well by plot (a). I'd suggest replacing the use of "map" with "flight" or "survey". I'd also suggest changing "secure" to "capture". Finally, how many hailstones are in this sample during flight 1?

Thanks for your advice to restrict the presentation to plot (a). The new title of this plot is now "'KDE evolution due to melting". Further we use the word "flight" instead of "map" and do not use "secure", but "capture".

The number of hail samples is summarized in Table 2 for all five surveys. During flight one we have 3925 hailstone samples for the melting analysis. This information is now included also in the caption of new Fig. 14 (old Fig. 12).

General comment:

I suspect that hail is most likely to fall such that the major and intermediate axes are visible from drone imagery. The minimum axis is most likely orientated to the vertical as the centre of mass is lowest to the ground at this most, and therefore has a (likely) high stability. This should be considered when discussing aspect ratio as a function of the major and minor axis.

Thanks for this very important comment, which is similar to a Referee #1 comment on the aspect ratios.

In Section 3.1 and the image caption of Fig. 9 (new Fig. 11), we now speak of projected aspect ratios:

"The projected hail aspect ratios indicate that the majority of hailstones have equal axis lengths (Fig. 11(b)) and 75% of the hailstones have projected aspect ratios higher than 0.75."

Further, we added a paragraph about the aspect ratios within the discussions (see answer to Referee #1 comment).

**References**

Kopp, J., Manzato, A., Hering, A., Germann, U., and Martius, O.: How observations from automatic hail sensors in Switzerland shed light on local hailfall duration and compare with hailpad measurements, Atmos. Meas. Tech., 16, 3487–3503, https://doi.org/10.5194/amt-16-3487-2023, 2023.

---

## Author Comment (AC2)

**Response to comment on amt-2023-89 by Anonymous Referee #1**

Summary: The authors report on drone-based measurements of hail sizes after an event in Switzerland that produced hail up to ~4 cm in maximum dimension. They describe a new, deep-learning based technique to automatically identify and size hailstones from the drone imagery. The technique is an improvement/extension (or at least a twist) on the methods reported in Soderholm et al. (2020, AMT). A particularly valuable contribution of this paper is the multiple drone missions to observe hailstone melting rates. The content is highly relevant to the hail community and is a timely contribution.

Unfortunately, I was disappointed to find at least 3 examples of plagiarism (see below) from websites in the description of some of the methods. I did not check for further instances, because the journal should have some ability to do so. In my view, plagiarism is a serious offense, and thus I recommend rejection at this time. However, I do find the research to be useful, and I do hope that the authors can rewrite the plagiarized portions of the manuscript in their own words, and address the other comments and suggestions below.

We very much appreciate the feedback by Referee #1 and all the raised comments, that help to improve the manuscript. All Referee comments are shown in black, the authors responses in green text color.

Major Comments:

1. I found at least 3 examples, based on where the style/tone of the writing abruptly changed. The first is on Lines 154-156. The text from the manuscript is as follows:

General answer:

Indeed, there are several instance in the manuscript where the text directly referred to an available source without correct citation. We admit that this should be clearly avoided in a scientific publication and want to thank the reviewers for pointing this out. We now carefully went through the manuscript to identify additional occurrences eventually not mentioned by the reviewers. All occurrences are related to definitions (e.g. an orthophoto, object detection) and specific characteristics of a software (e.g. OpenSFM library) where the text from a very early version of the draft unfortunately remained until submission. We now rephrased all relevant sentences and wrote it in our own words. In addition we directly refer to the relevant resources from with the information is taken.

"An orthomosaic is a photogrammetrically orthorectified image product that has been mosaicked from an image collection, correcting for geometric distortion and color matching the image data to create a seamless mosaic data set."

and from the ArcGIS website (https://pro.arcgis.com/en/pro-app/latest/help/data/imagery/generate-an-orthomosaics-using-the-orthomosaic-wizard.htm#:~:text=An%20orthomosaic%20is%20a%20photogrammetrically,produce%20a%20seamless%20mosaic%20dataset):

"An orthomosaic is a photogrammetrically orthorectified image product mosaicked from an image collection, where the geometric distortion has been corrected and the imagery has been color balanced to produce a seamless mosaic dataset."

The corresponding lines are now adapted and the definition of an orthophoto is now written in our own words.

Line 129-133 in the revised manuscript now reads: "An orthomosaic is defined as a composite of multiple aerial (airborne or space-borne) photos that are previously processed to remove inherent distortions caused by the geometrical properties of the lenses (airborne photos) and the earths curvature (space borne satellite images). Thus, the processed individual pictures and the resulting composed orthomosaic is distortion free and exhibits a true scale that allows to estimate the size of the objects within the photo."

The second is on Lines 168-169. The text from the manuscript:

"The library serves as a processing pipeline for reconstructing camera poses and 3-dimensional scenes from multiple images. Here we make use of some basic modules for SfM: Feature detection, feature matching, minimal solvers."

is largely taken from the github page for this software (https://github.com/mapillary/OpenSfM/blob/main/README.md):

"OpenSfM is a Structure from Motion library written in Python. The library serves as a processing pipeline for reconstructing camera poses and 3D scenes from multiple images. It consists of basic modules for Structure from Motion (feature detection/matching, minimal solvers) with a focus on building a robust and scalable reconstruction pipeline."

The corresponding lines are now adapted and rephrased in our own words. In addition, we refer to the official github page of this software.

Line 143-145 in the revised manuscript now reads: "The library can be used to reconstruct camera positions and 3-dimensional scenes based on multiple images (mapillary, 2023). Here we make use of the basic modules for SfM: Feature detection, feature matching, minimal solvers."

The third is from Lines 192-193. The text from the manuscript:

"Object detection is a technology related to computer vision and image processing that tries to detect instances of semantic objects of a certain class (e.g. cats, dogs, cars, buildings, etc.) in digital images and videos."

is taken from the following website (https://www.credly.com/skills/image-processing-object-detection#:~:text=Object%20detection%20is%20a%20computer,in%20digital%20images%20and%20videos.):

"Object detection is a computer technology related to computer vision and image processing that deals with detecting instances of semantic objects of a certain class (such as humans, buildings, or cars) in digital images and videos."

The corresponding lines are now adapted and rephrased in our own words.

Line 165-166 in the revised manuscript now reads: "Object detection is a computational method to automatically identify and locate different objects or semantic classes (e.g. trees, bicycles, faces) within an image or a video."

2. Section 2.1: I appreciate the detailed information and experiences in this section, but it comes across as a little bit "preachy" or reads like pontification. Please take a look at this section and try to trim it down to what is necessary and germane for the main story about the new technique.

We now shortened this section to avoid any "preachy" character related to our experience. In particular, we deleted the paragraph highlighting the dangers associated with chasing thunderstorms and focus on specific technical aspects that are important to collect successfully collect data, which can be analyzed using the proposed methods.

3. How would this motion blur affect the hail size results? This should at least be mentioned here, even if the answer is "not at all" so you do not leave the readers wondering.

Motion blur would lead to small (< 1.5 mm) overestimations of the hail dimensions, as we use camera settings and flight speeds to reduce the motion blur to below one pixel size (1.5 mm).

We will clarify this in Section 2.2:

"A low horizontal flight speed is necessary to reduce the motion blur (Bemis et al., 2014; Soderholm et al., 2020), which is within one image pixel in our case and leads in general to small overestimations (< 1.5mm) of the hail dimensions."

4. Lines 339-340: The authors should make a note here that the aspect ratios reported are probably not the same as measured in other studies (Knight 1986, Shedd et al. 2021), which are the measured maximum and minimum axes of the hailstones. What the drone sees are the projected maximum and minimum axes, based on whichever way the hailstone happens to be laying on the field. If hailstones are perfectly oblate spheroids, you would always capture the maximum dimension but not always the minimum dimension. Because hailstones tend to be ellipsoidal or irregular, this means your axis ratios probably do not correspond to the true stone axis ratios.

Thank you for this important comment on the aspect ratios and your clarifications, which we take partly into our discussions.

In Section 3.1 and the image caption of Fig. 9 (new Fig. 11), we now speak of projected aspect ratios:

"The projected hail aspect ratios indicate that the majority of hailstones have equal axis lengths (Fig. 11(b)) and 75% of the hailstones have projected aspect ratios higher than 0.75."

Further, we added and expanded a paragraph (information moved from introduction) about the aspect ratios within the discussions:

"Hailstones usually have an oblate spheroid shape with mean axis ratios close to 0.8, though they can sometimes have large protuberances (Knight, 1986) and the probability for nonspherical shapes rises with increasing maximum dimension (Shedd et al., 2021). As a consequence the hail aspect

ratio decreases for larger sizes as shown in the various studied data sets (Knight, 1986; Soderholm et al., 2020; Shedd et al., 2021). Figure 6 in Shedd et al. (2021) compares their recent results on the evolution of aspect ratios with maximum hail dimensions from manually measured hailstones to the results of Knight (1986). The slopes of the decreasing aspect ratios are comparable, but the absolute values tend to be lower in the hail data set of Shedd et al. (2021), reflecting possible effects by melting before the measurements were taken. Likewise with hailpads, the shape factor in the image plane can be determined with the aerial drone-based hail photogrammetry, but the estimated aspect ratios (Fig. 11(b)) may differ from in-situ measurements as published in e.g. Knight (1986); Shedd et al. (2021). The hail images show only the projected maximum and minimum axes, which may differ to the true stone axis ratios."

Minor comments/Typos/Grammar issues:

1. Line 24: "asses" should be "assess"

The typo has been corrected.

2. Line 29: I think "alps" should be capitalized? Same in Line 60?

Thanks, we now capitalized all instances of "alps".

3. Line 39: probably more accurate or clearer to say "maximum dimension" instead of "diameter" (the latter connotes a sphere or circle)

In general we now do not use the term "diameter" anymore and rather speak of "dimension" or simply "size".

4. Line 52: no comma after "known"

The comma after "known" is deleted now.

5. Line 53: A more comprehensive and more recent study is by Shedd et al. (2021, JAS) that looks at hailstone shapes; consider comparing the Knight (1986) results to those of Shedd et al. here.

Thank you for mentioning the Shedd et al. (2021) study, which we now included for a brief comparison to Knight (1986) within the discussion section. See also our answer to your previous comment on lines 339-340.

6. Line 54: Soderholm et al. (2020, AMT) also report on the axis ratios, correct?

Yes of course, now in this paragraph within the discussions (see previous comment), we added the citation of Soderholm et al. (2020) as well.

7. Line 56: "decent" is a bit informal, is there a way to quantify what this means?

Now we omit the word decent and reformulated the text passage to:

"That day, the ingredients for long-living and well-organized severe thunderstorms (humid air, high instability and strong wind shear) were in place across Switzerland. An air mass with steep lapse rates was advected from the southwest above a moist low-level air with mean mixing ratios around 12 g kg$^{-1}$."

The mixing ratio value has been derived from the Payerne sounding at 12UTC.

8. Line 61: what is the lowercase s? Is this South? If so, it might be clearer to spell it out. Update, it happens again in Line 62, so I don't know what this means. Please spell it out.

The "s." should have been the abbreviation of the word "see". E.g. …(see Fig. 1).. etc. There are many more instances of this abbreviation in the manuscript. Now we always write the full word "see".

9. Line 74: "respectively" is used incorrectly here, should read as "at a distance of 770 m and 1470 m, respectively, to the NNE of…"

Thanks, this instance of respectively in line 74 has been corrected.

10. Figure 1, caption: the description in the caption regarding EMLs and "loaded gun" belongs in the text. However, "loaded gun" is a bit colloquial, consider using other terminology. Check on the convention for how to portray units (i.e., m/s or m s^-1, etc.) for AMT. Finally, explain or provide a legend for what the colors mean in the hodograph, and indicate the units (m/s or kts?) for the rings on the hodograph.

Now Figure 1 is revised in the way, that we reduced the number of barbs (Referee #2 comment) and we removed any interpretation from the caption (no use of "loaded-gun" anymore). All units styles should be AMT conform as well. We changed the color for the layers in the hodograph (0, 1, 3, 5, 10 km) and they are defined now in the caption of Fig. 1 . Further the label on the hodograph axis was added and we changed from m/s to kt units. The MetPy software reference has been added and please note there is a slight change in the absolute values for e.g. CAPE, CIN, SRH likely due to the use of a newer version of MetPy.

11. Figure 2: consider enlarging the dots for the hail reports, they are all very small and hard to see.

Thanks for this suggestion, we enlarged the markers for the hail reports (by 20%) and hope the visibility improved. For further adjustments on this Figure 2, please refer to the corresponding comment and answer to Referee #2.

12. Section 2: the first 3 or 4 lines are probably not needed, since they are just telling readers what is coming up. How about just start with the material? Similarly, the second sentence of subsection 2.1 can be removed, it is useless for the narrative of the paper.

Yes you are absolutely right, the first sentences here are not strictly needed. Now we skip these sentences. Also the second sentence of subsection 2.1 is removed now. Please note that substantial language and text editing have taken place here too (Referee #2 comment).

13. Line 106: no comma needed after "found"

Due to substantial language and text editing here, the start of the mentioned sentence with "It was found, …" is different now.

14. Line 113: "Aside the" should be "Aside from the"

Now we write "Aside from the …" in line 99 (revised manuscript).

15. Lines 119-121 are not needed – it is pretty obvious that any field experiment would require good forecasting ahead of time! Just start with "During days with conditions favorable for supercells," or something like that.

We now start the sentence (line 104) as suggested with:

"During days with conditions favorable for supercells, the drone operators…".

16. Line 130 and elsewhere, "hereby" is not the correct word to use here. Please revise.

In the whole manuscript, we do not use the word "hereby" anymore.

17. Line 136: "hail core punch" is too colloquial, please revise.

Section 2.1 has undergone substantial language changes and we do not use the expression "hail core punch" anymore.

18. Line 141: "analyses" should be "analysis"

At the end of Section 2.1 we now write:

"… for an in depth analysis."

19. Lines 142-147: Even though these are important points for storm chasers, I don't think these are appropriate for the manuscript because they aren't relevant for reporting on the technique and results. Please remove.

Like suggested, this paragraph at the end of Section 2.1 (old version) is removed now.

20. Line 206: no comma after "mentioned" (And, if you're writing it in the paper, it seems worthy of mentioning. Best practice is to not include text like "It is worth mentioning" etc. and just cut to the chase with the important points).

Thanks for pointing this out. Substantial language, text editing and reformulations have taken place and now we try to avoid such expressions.

21. Line 231: "tow" should be "two"

Corrected now.

22. Line 241: the brackets usage for quotes here needs to be changed to conform to AMT's convention/guidelines. This occurs throughout the manuscript.

We now avoid these quote signs within brackets («») and write the e.g. run-3 instances in italic font instead. In case it is not AMT confirm yet, it will be corrected during official typesetting.

23. Lines 354-355: This is repeating the finding from the first sentence in the paragraph; combine these two sentences into one and keep them together in the text (otherwise the logic is jumping around).

Yes, we agree that this should be improved. The whole readability in this part of the manuscript should be better now. The second paragraph of Section 3.2 starts with:

"Within all virtual hail sensors only 45 hailstones with a size larger than 30mm are observed and thus only 0.3% (34 out of 10000) of the virtual sensors exhibit an impact of such large hail."

24. Lines 356-367: There are several one-sentence paragraphs here; simply combine them into a coherent paragraph with connecting sentences or words.

Section 3.2 now consists of only 3 paragraphs.

25. Line 377: Does the changing shapes of the larger hailstone agree with the cartoon drawn in Shedd et al. (2021)? In other words, is there evidence that protuberances or lobes melt more rapidly than the rest of the stone, tending to "smooth" the stones?

Here we have to mention, that based on visual observations on site and also the drone images, the hail stones were not showing extensive protuberances with this event. Even if the statement

("protuberances melt more rapidly") seems to be true for the stone 1 (top row in Fig. 11), we are not able to make a statistically profound statement about it yet.

Further we would need to put much more effort into the validation of the mask shapes, if they capture also the small lobes. For future studies this is for sure a very interesting point of investigation.

26. Line 387: No comma needed after "range" (or after "bins" on the next line)

Due to substantial language and text editing here, the wording changed now.

27. Line 389: But, certainly, physics tells us that there should be some dependence on size, right? One can refer to Rasmussen and Heymsfield (1987, JAS), for example.

We agree with the referee that the physics describing the melting of ice particles could be briefly mentioned. We added the following information from the mentioned  reference within the Section 4 (Discussion) lines 373-378:

"Other studies by Rasmussen and Pruppacher (1982) and Rasmussen and Heymsfield (1987) have explored the melting of spherical ice particles falling at terminal velocity. They found that the melting rate depends on the initial size of the spheres size and the surroundings, including temperature, humidity, turbulence, and how meltwater is shed. The hailstones in our case are already on the ground, so they experience different environmental conditions compared to when they are falling through the atmosphere. We have not measured these specific conditions for each hailstone, so we cannot make any conclusions about how the melting rate relates to their initial size."

28. Line 394: This is important information that could be included earlier in the text, near the description of the event!

In the introduction following line 60 (revised manuscript version) we include now the on-site observations:

"For this location MESHS indicates a maximum expected severe hail size of 63mm and on-site observations revealed maximum dimensions between 40mm and 50mm."

---

## Referee Report (RR1)

Review of "Drone-based photogrammetry combined with deep-learning to estimate hail size distributions and melting of hail on the ground"

Summary:
This manuscript presents a case study of using drone-based photogrammetry and deep learning to identify and classify hail size distributions over a soccer pitch in Switzerland. The technique is an advancement of Soderholm et al. (2020) and is a promising way to determine hail size distributions, including the effects of melting, from hail swaths on the ground. The authors compare their results to automatic force-detection hail sensors, radar-based Maximum Expected Severe Hail Size measurements, and a subset of expert evaluations. The manuscript is well-written overall.

Major Comments:

1. I am concerned about the reliability of the small (<6 mm) hail measurements, and I think it would be good for the authors to more directly address and/or plan future follow-ups. These are
   a. ISO 25,600, while not as problematic on modern full-frame camera sensors as in the past, still produces quite a bit of noise. When examining areas on the order of 1-4 pixels, as would be required for hail sizes below 6 mm, areas of noise could very easily be identified as hail. How was mitigation performed?

   b. The authors briefly discuss the impact of motion blur, but for small hail sizes, it could make a larger impact than the authors say. A 1/1000 shutter speed with the drone moving at 1.5 m/s would indicate to me that a single 1.5 mm hailstone could be "smeared" across two neighboring pixels, appearing as a single 3 mm hailstone.

   c. To be clear, the values for shutter speed and ISO are reasonable, and the authors discuss the challenges of lack of light. However, more discussion and/or validation at the image collection step in the manuscript would enhance it, which is otherwise not accounted for.

2. I did not see much discussion on how the aspect ratios were determined. I am particularly concerned about the quality of aspect ratio measurements for small hail sizes; I'm a bit perplexed as to how the aspect ratio for small hail is determined given the relatively coarse pixel size versus hail size.

Minor Comments:

1. There are several minor grammatical and/or punctuation issues in the manuscript, but I will defer to the copywriting staff to identify and resolve.
2. Section 2.1: This section feels too long and not as relevant to the rest of the manuscript.
3. Line 152: it would be good to note the temperature in here.

4. Line 162: How much smaller are the black circles? How does this impact the measurements?
5. Line 162: Is there a reason that the overexposure was not corrected for after the fact? Were any highlights in the pictures clipped?
6. Line 196: If the additional experts are annotating the same validation and test data as expert A, I'm not sure that these can be described purely as independent comparisons for the ML model.
7. Line 205: the trademark symbol feels unnecessary
8. Section 3.3: What is the accuracy of the orthophotos? I am concerned that the hail pixels are moving substantially enough that a 1:1 comparison in hail stone size isn't possible.

---

## Referee Report (RR2)

AMT-2023-89 Report

This research represents a somewhat incremental but important step in advancing hail estimates and provides innovative development to address several challenges and make improvements. While building on the work of HailPixel (Soderholm et al., 2020), the authors' key contributions are: (1) a demonstration that the image processing pipeline can be reduced to only applying a region-based convolutional neural network (R-CNN) to an orthomosaic, both simplifying the processing and increasing the accuracy, and (2) an analysis of hailstone melting rates, which is critical to accurately understand any post-event hail observations, using successive flights. Furthermore, the data collection procedures presented continue to refine strategies for the successful interception and observation of hail events, which is non-trivial. There are some details of procedures and implementation that are not fully optimized but seem within reason for initial experimental purposes. Overall, the greatest limitation of this work is the very limited data set (a single event), which makes it difficult to understand the broad applicability of some specifics (for example, the R-CNN model as-trained or loss of accuracy due to lighting conditions). However, it represents valuable proof-of-concept with novel approaches and would serve as a steppingstone for future work. The techniques applied lend themselves easily to such future development and expanded data collection, relying only on commercial-off-the-shelf equipment and consumer-grade computing equipment.

Comments/Questions/Suggestions:

1. Line 56 "lake" -> "Lake"

2. Line 62 "for the area within a distance of less than 1 km from the survey area" -> "for the area within 1 km of the survey area"

3. Lines 69-70 "a high resolution" is ambiguous, what is the resolution? (It is given on Line 126.)
   Similarly, Line 53 "giving a ground sampling distance (GSD) of 1.5 mm px−1" From what altitude of flight? (It is given on Line 137.)
   It would be nice to have all of this information presented in a single statement or at least the same section (e.g. something like "A ground sampling distance (GSD) of 1.5 mm px−1 was achieved flying at an altitude of 12m with a 45 MP camera."), similar to how it appears in your abstract. Perhaps even a flight characteristics table would make referencing this comparatively in future work easier. This is not a critical point as all the information is presented, but as a matter of preference could be easier to consume.

4. Line 91 "better", Is it possible to present a quantified comparison for this?

5. Section 2.2. Did you have any guidelines for acceptable flight conditions? E.g. Maximum wind speed/gusts, etc. It is clear you were trying to get off the ground as soon as possible

after an event, but given the criticality of timing, it would be helpful to know if there were any additional limitations.

6. Line 154 "ISO-25600" This is very high and likely introduces a fair bit of noise. This is explained later on Line 428, but I think it would be useful to include the reasoning for such a high ISO in the earlier section. Could results be improved by running a slightly slower shutter and lower ISO? Is wind playing a role in image blur in addition to forward flight? These may be topics for future work, but it would be nice to see them acknowledged if applicable.

7. Line 157 GPS Error: Are you using RTK correction? The error value suggests not, even though your drone supports it, and you call this out explicitly (Line 127). There's obviously a substantial challenge in deploying an RTK base station and establishing a usable dilution of precision in the timeframes you require. However, it would be worth mentioning these limitations and maybe potential alternatives, such as NTRIP (Networked Transport of RTCM via Internet Protocol) services, if available, especially given the discussion in Section 3.3 of lacking positional consistency without ground references such as the soccer center circle.

8. Line 297 "equal axis lengths". From your bin definitions, this actually represents aspect ratios >0.9, not necessarily exactly equal.

9. Figure 5(d) doesn't make sense until you get to Line 300. I understand why it makes sense to have with (a)-(c) as a single figure, but it may be worth noting the section it applies to in the caption.

10. Lines 306-309 might be more easily digested as a table, but that's more a preference.

11. Section 4 – First paragraph could maybe go in intro, feels a little out of place here, but again more of a preference.

12. Line 367 "as published in e.g. Knight (1986); Shedd et al. (2021)" -> "as published in Knight..."

13. Table 3 – Would it be possible to make this as a time series plot, T on left axis, RH on right (or similar)? It is more difficult to pick out the trends looking at a table. Noting the flight times as vertical lines or highlighted sections would further help in understanding the overall timeline of events.

14. Line 385 "what might effect" -> "which might affect"

Overall largest concern: Is this repeatable and generalizable? Your results are based on a single event used for training, validation and testing. Very interesting work though and sets the stage for future research that can begin to fine tune and hopefully more extensively validate these types of analyses.

Other suggestions for potential future research:
(No expectation of these for this publication but curiosities that may be of interest to the authors.)

- Integration of thermal imagery. Even with low resolution, the integrated pixel values could provide useful information. By using the surface temp in areas with high probability and confidence of not having hail present as a background, you could use the differentials of other pixels to help include or exclude potential hailstones in conjunction with RGB techniques.
- Utilizing SfM result and applying R-CNN directly to mesh or point cloud rather than 2D orthomosaic. This would obviously require more computing power, but it would be interesting to see how it changes performance in 2D visually challenging environments (like taller grass).

---

## Author Response (AR2)

**Response to comment on amt-2023-89 by Anonymous Referee #2**

Thanks for your detailed responses and considerable effort to improve the overall quality of the text. It's in excellent shape and I recommend accept with minor technical corrections:

Comment on final figure / line 385 (of original manuscript): Thanks for following up on my poor example, I agree that it's hard to convey this information.

I think the existing plot is suitable given the limitation of not tracking the melting of all individual hailstones.

We highly appreciate, that Referee #2 was available for the second round of review of our manuscript. We agree, that the current representation of the melting process of hailstones in Figure 14 is sufficient. Further, we note that with this minor revision we also make the hail data collection for the given event publicly available (https://doi.org/10.5281/zenodo.10609730) for future analyses.

Figure 2: I can't see the following in the track dots: "The storm location at the sounding time (12 UTC) is marked with the same edge color (magenta)" In (2) I can only see a black rectangle and white cross, no white rectangle and magenta cross.

Unfortunately we forgot to adapt the caption of Figure 2 regarding the black rectangle and white cross. We correct this and add also the magenta edge color marker in subplot (a) again, originally it was there. The resolution of the base map (terrain-background) of Figure 2(b) has been increased, as it was quite pixelated. As raised by the editorial staff, we add copyright statements about the used map material to this Figure: "Map tiles by Stamen Design (stamen.com) and Stadia Maps (stadiamaps.com), under CC BY 4.0. Map data copyrighted by OpenStreetMap contributors and available from https://www.openstreetmap.org." In the Figure itself, we additionally give attribution to the Natural Earth data (https://naturalearthdata.com) with an official logo.

Figure 3: HS1 is really hard to spot, took me a full minute, maybe change the color?

Thank you for sharing this experience. Of course we can change the color of HS1 marker. We will use yellow color for HS1 now and add black edge lines around the markers for better visibility.
As raised by the editorial staff, we add copyright statements about the used map material to this Figure: "Map tiles by Stamen Design (stamen.com) and Stadia Maps (stadiamaps.com), under CC BY 4.0. Map data copyrighted by OpenStreetMap contributors and available from https://www.openstreetmap.org."
To stay compliant with the hail sensor color scheme, the cyan line Figure 12(c) is now yellow.

Figure 5: I can now see why you preferred to use the virtual sensor area instead of center location. Please keep as it is. Thanks for checking this.

Thank you for your comment, we will keep Figure 5(d) as it is.

**Response to comment on amt-2023-89 by Anonymous Referee #3**

This manuscript presents a case study of using drone-based photogrammetry and deep learning to identify and classify hail size distributions over a soccer pitch in Switzerland. The technique is an advancement of Soderholm et al. (2020) and is a promising way to determine hail size distributions, including the effects of melting, from hail swaths on the ground. The authors compare their results to automatic force-detection hail sensors, radar-based Maximum Expected Severe Hail Size measurements, and a subset of expert evaluations. The manuscript is well-written overall.

We thank Referee #3 for his feedback and comments to further improve our manuscript. Please find our point-by-point answers to these comments below.

Major Comments:
1. I am concerned about the reliability of the small (<6 mm) hail measurements, and I think it would be good for the authors to more directly address and/or plan future follow-ups. These are
   a. ISO 25,600, while not as problematic on modern full-frame camera sensors as in the past, still produces quite a bit of noise. When examining areas on the order of 1-4 pixels, as would be required for hail sizes below 6 mm, areas of noise could very easily be identified as hail. How was mitigation performed?

      So far, no mitigation was performed and a potential solution could be to introduce artificial illumination as discussed in Section 5. There are plans to work on this issue for future data collections.

   b. The authors briefly discuss the impact of motion blur, but for small hail sizes, it could make a larger impact than the authors say. A 1/1000 shutter speed with the drone moving at 1.5 m/s would indicate to me that a single 1.5 mm hailstone could be "smeared" across two neighboring pixels, appearing as a single 3 mm hailstone.

      The drone was flying at a speed of 1 m/s and thus the exact value for motion blur is 0.67 pixels with a GSD of 1.5mm/px. We looked a bit closer to the small detected hail stones and found that there are only 2 classified between 1 mm and 2 mm. Those were not included in our analyses of the HSD, where the smallest bin size was between 3 mm and 6 mm. Thus we correct the total number of hail stones found to 18207 in the text and plot titles.

      In lines 138 and 139 we more clearly specify the motion blur value and expected overestimations:
      "…, which is within one image pixel (0.67 px) in our case and leads in general to small overestimations (≈ 1 mm) of the hail dimensions."

      We agree, that the highest uncertainties are present for the small hailstones. The hail size classes below 1 cm suffer most from the blur effect, which might not be neglectable. Furthermore, wind induced motions of the drone might introduce additional blur. This circumstance is added in Section 2.2:

      "The utilization of a relatively high ISO value, as outlined in Table 1, facilitates operational use even in challenging lighting conditions, maintaining low motion blur

(0.67 px) at a constant flight speed of the drone. Furthermore, wind and gusts can affect the drone's stability, potentially leading to additional image blurring."

The small hail size classes should be further assessed in future analyses. However, hail size below 1 cm does not have a large damage potential and thus we focus on larger size classes. As a concept of proof we currently keep all size classes but in future analyses this effect should be further investigated.

   c. To be clear, the values for shutter speed and ISO are reasonable, and the authors discuss the challenges of lack of light. However, more discussion and/or validation at the image collection step in the manuscript would enhance it, which is otherwise not accounted for.

The image collection is one of the biggest challenge in drone based hail observations. This means within extremely short time and under extreme environmental conditions, which cannot be reproduced in a training, the collection must take place in the first try. Thus we designed the collection to be successful within as much illumination ranges as possible. However, various plans to design synthetic experiments for validation of different light conditions should be taken into account for future analyses and are currently planned for our upcoming data collections.

2. I did not see much discussion on how the aspect ratios were determined. I am particularly concerned about the quality of aspect ratio measurements for small hail sizes; I'm a bit perplexed as to how the aspect ratio for small hail is determined given the relatively coarse pixel size versus hail size.

We agree that the quality of the aspect ration might be a concern, as mentioned above regarding the motion blur effecting small hail sizes, future studies should consider a separation of size classes. However, this is out of scope of the current study, but still the results ,e.g. mean axis ratios close to 0.8, are in agreement to other studies like Knight (1986). Despite these quality limitation, there is hardly an alternative to analyze axis ratios of large amounts of real world hailstones. Hence, based on our results, further investigations and an effort to increase the quality should be taken into account for future studies.

Minor Comments:

1. There are several minor grammatical and/or punctuation issues in the manuscript, but I will defer to the copywriting staff to identify and resolve.

Thanks for spotting these grammatical/punctuation issues. We will walk through the text carefully again and the copy-editing stage will hopefully correct the remaining errors.

2. Section 2.1: This section feels too long and not as relevant to the rest of the manuscript.

We note, that this section was already substantially shortened after the first revision and we would like to keep it as it is to share some experiences with the reader. It might also help top plan the data collection process for future field campaigns in particular in complex terrain.

3. Line 152: it would be good to note the temperature in here.

In principle, this is a good idea, but we lack temperature measurements directly at the hail survey site in Entlebuch. The nearest SMN station, located in Schüpfheim (Table 3), is 5.7 km away. We believe it is more appropriate to address the temperature discussion in Section 4.

A comment by Referee #4 brings a change. Table 3 has been replaced with a more easily digestible plot, which also displays the start time of the hailfall and the various flight times required to cover the soccer middle circle area.

4. Line 162: How much smaller are the black circles? How does this impact the measurements?

Thank you for this comment. The black circles against white background appeared approximately 1-2 pixel smaller than the white ones against the black background, likely due to overexposure. This finding does not affect the hail size estimation, as hail is a bright object. We now more specifically write:

"Due to a slight overexposure in combination with the motion blur, the black circles on white background appeared approximately 1-2 pixels smaller."

5. Line 162: Is there a reason that the overexposure was not corrected for after the fact? Were any highlights in the pictures clipped?

Because the effect had only a remarkable influence on dark objects against a bright background and not vice versa, we did not consider a correction for overexposure. And no, we did not clip any "highlights". For the analyses part, e.g. the reference objects were masked in black color in order to prevent a false classification of the white circles as hail. In general to ease the image correction for future captures of hail, we strongly recommend to save the RAW images as well, which was unfortunately not the case for the presented event.

6. Line 196: If the additional experts are annotating the same validation and test data as expert A, I'm not sure that these can be described purely as independent comparisons for the ML model.

The additional human experts (B and C) annotated only the same test dataset, which was not utilized for tuning the model. In this sense, we consider it an independent comparison to assess the ML model results. However, our intention was also to highlight the differences present in the annotations of the experts. It shows another source of uncertainty that is present within the whole process.

7. Line 205: the trademark symbol feels unnecessary

Thanks, yes this we will remove.

8. Section 3.3: What is the accuracy of the orthophotos? I am concerned that the hail pixels are moving substantially enough that a 1:1 comparison in hail stone size isn't possible.

As given in Section 2.2, Line 157: the mean GPS error of the first drone flight is 0.34 m. This varies between the flights 1-5 between 0.21 m and 0.5 m. This circumstance makes a simple 1:1 comparison of the hailstone positions impossible. This we clearly state in Lines 323-325. However this does not affect the GSD on the ground which stays (confirmed by the

reference objects) at 1.5mm/px. Thus the alteration of the distributions from the 5 flights within the soccer middle circle can be well attributed to the melting process.

Regarding this we think it is worthwhile to clearly mention this in Section 3.3:
"Due to slight deviations in the derived orthophotos from varying GPS errors (0.21 m to 0.5 m) and the melting process itself,…"

And we add at the end of the paragraph:
"The GSD between the flights stays constant at 1.5 mm/px, as confirmed by the reference objects."

**Response to comment on amt-2023-89 by Anonymous Referee #4**

This research represents a somewhat incremental but important step in advancing hail estimates and provides innovative development to address several challenges and make improvements. While building on the work of HailPixel (Soderholm et al., 2020), the authors' key contributions are: (1) a demonstration that the image processing pipeline can be reduced to only applying a region-based convolutional neural network (R-CNN) to an orthomosaic, both simplifying the processing and increasing the accuracy, and (2) an analysis of hailstone melting rates, which is critical to accurately understand any post-event hail observations, using successive flights. Furthermore, the data collection procedures presented continue to refine strategies for the successful interception and observation of hail events, which is non-trivial. There are some details of procedures and implementation that are not fully optimized but seem within reason for initial experimental purposes. Overall, the greatest limitation of this work is the very limited data set (a single event), which makes it difficult to understand the broad applicability of some specifics (for example, the R-CNN model as-trained or loss of accuracy due to lighting conditions). However, it represents valuable proof-of-concept with novel approaches and would serve as a steppingstone for future work. The techniques applied lend themselves easily to such future development and expanded data collection, relying only on commercial-off-the-shelf equipment and consumer-grade computing equipment.

We thank Referee #4 for his positive evaluation and comments to further improve our manuscript. Your observation is indeed right. We perceive this contribution as a continuation following the HailPixel research by Soderholm et al. (2020). Subsequent advancements are anticipated to build upon this foundation. And with more captured hail events, the techniques to identify hailstones and calculate the HSD can be finetuned, and hopefully more generalized at some point.

Comments/Questions/Suggestions:
1. Line 56 "lake" -> "Lake"

   Typo has been corrected on Line 56.

2. Line 62 "for the area within a distance of less than 1 km from the survey area" -> "for the area within 1 km of the survey area"

   We simplified the sentence as suggested.

3. Lines 69-70 "a high resolution" is ambiguous, what is the resolution? (It is given on Line 126.)

   Similarly, Line 53 "giving a ground sampling distance (GSD) of 1.5 mm px−1" From what altitude of flight? (It is given on Line 137.)

   It would be nice to have all of this information presented in a single statement or at least the same section (e.g. something like "A ground sampling distance (GSD) of 1.5 mm px−1 was achieved flying at an altitude of 12m with a 45 MP camera."), similar to how it appears in your abstract. Perhaps even a flight characteristics table would make referencing this comparatively in future work easier. This is not a critical point as all the information is presented, but as a matter of preference could be easier to consume.

We agree that it is useful for the reader to have these information altogether in one sentence. In Lines 69-70 we now specify:

"Following those suggestions, we achieved a ground sampling distance (GSD) of 1.5 mm/px by flying at an altitude of 12 m with a 45 megapixel full frame camera system. For the detailed flight and system characteristics see also Table 1."

In Section 2.2 we incorporated a new table (Table 1) to summarize the drone, camera and flight characteristics of the main mission ("Specifications of the drone, camera system and flight characteristics."). Thus we shortened the belonging text parts a bit by linking to the information in Table 1.:

"Table 1 summarizes the detailed drone, camera system and flight characteristics of the hail photogrammetry mission."

4. Line 91 "better", Is it possible to present a quantified comparison for this?

Sorry for not being very precise in our formulation. We now specifically say:

"With the two-stage approach, the edge detection did not work reliable in particular for small hail stones, because the lightness gradient between these small hail stones and the background was insufficient. Here we therefore focus on the one-stage approach."

A more quantitative and in depth analyses of the performance of edge-detection algorithms for small hailstone sizes will for the moment be left to future research. This in general is also connected to image quality, noise and motion blur. Here we just show a simple example of a small hailstone (left image), where our edge-detection (right image) obviously failed:

[Figure]

The segmentation mask from the R-CNN model (middle image) is much closer to the true stone shape.

5. Section 2.2. Did you have any guidelines for acceptable flight conditions? E.g. Maximum wind speed/gusts, etc. It is clear you were trying to get off the ground as soon as possible after an event, but given the criticality of timing, it would be helpful to know if there were any additional limitations.

Although the Matrice 300 reaches IP45 water protection, there is no product warranty to cover water damages. It is the risk of the operator when flying in wet conditions. We followed our defined rule to not fly the drone if the rain rate exceeds 4-5 mm/h. Safe operation is guaranteed up to a wind and gust speed of 10 - 15 m/s. These values of course depend on the drone system in use.

In Section 2.2 we suggest to add the following sentence: "Although it is critical to get off the ground as soon as possible after the hailfall, environmental conditions like rain rate, wind and gust speed should be carefully watched out in order to stay within the permitted operation conditions of the drone model."

6. Line 154 "ISO-25600" This is very high and likely introduces a fair bit of noise. This is explained later on Line 428, but I think it would be useful to include the reasoning for such a high ISO in the earlier section. Could results be improved by running a slightly slower shutter and lower ISO? Is wind playing a role in image blur in addition to forward flight? These may be topics for future work, but it would be nice to see them acknowledged if applicable.

We think that much more field tests are needed to find the optimal settings for the camera dependent on the available light conditions. Sensitivity tests need to be performed and could be another step towards future improvements of the method. There is also the possibility to install a light source next to the camera with a dual-gimbal mounting device, which could be very useful to reduce the ISO by keeping a fast shutter speed. We discuss the potential of a light source and state that the image quality can be improved by reducing the ISO in Section 5. We also included a point in the new list of ideas:

- Fine tuning of hardware settings and flight characteristics for optimal image quality in conjunction with an acceptable motion blur.

The reasoning for the high ISO is added now in the earlier Section 2.2:
"The usage of a quite high ISO (see Table 1) value is explained by simplifying the operational usage also in difficult light conditions, while keeping the motion blur low (0.67 px) at a constant flight speed of the drone. …"

Yes, wind and especially gusts can reduce the stability of the drone flight and thus have the potential to add additional image blur. But it is really difficult to quantify the contribution of wind to the blur effect and is currently out of scope for this work.

To the paragraph above (within Section 2.2) we add: "In addition, wind and gusts may influence the stability of the drone movement, which could further blur the image."

Regarding image quality it would be best to program the drone to stop at each capture point to remove the image blur from forward motion. However the amount of time then would largely increase and lower the size of the area that can be mapped. There are tradeoffs one has to decide on.

7. Line 157 GPS Error: Are you using RTK correction? The error value suggests not, even though your drone supports it, and you call this out explicitly (Line 127). There's obviously a substantial challenge in deploying an RTK base station and establishing a usable dilution of precision in the timeframes you require. However, it would be worth mentioning these limitations and maybe potential alternatives, such as NTRIP (Networked Transport of RTCM via Internet Protocol) services, if available, especially given the discussion in Section 3.3 of lacking positional consistency without ground references such as the soccer center circle.

Thank you very much for this comment. You are right we called the RTK feature out, but actually did not operate the drone with an RTK base station. As you say, it introduces a lot of additional efforts. We will clarify this in the text also with regard to the positional inconsistency between the flights.

In line 127 we now do not mention the RTK feature and clarify thereafter:

"The drone was not operated with enabled RTK (Real Time Kinematic) feature. This would require the installation of a RTK base station module. The advantage would be an increase in positional accuracy of the drone from the order of few decimeters to centimeters. Another potential option would be to use the NTRIP (Networked Transport of RTCM (Radio Technical Commission for Maritime Services) via Internet Protocol) protocol. This protocol facilitates the transmission of correction data over the internet. It enables real-time positioning and precise navigation by delivering accurate correction data to GPS receivers."

In Section 3.3 , also with respect to a comment by Referee #3, we specify the GPS errors more detailed now:  "…from varying GPS errors (0.21 m to 0.5 m) …"

We note, that we unified the appearance of the expressions "soccer middle circle" and "soccer center circle" to "soccer middle circle" in the whole text.

8.  Line 297 "equal axis lengths". From your bin definitions, this actually represents aspect ratios >0.9, not necessarily exactly equal.

    Thanks for spotting this, here we change the formulation now to: "The projected hail aspect ratios indicate that the majority of hailstones are rather spherical with axis ratios greater 0.9 (Fig. 11(b). 75 % of the hailstones have projected aspect ratios higher than 0.75."

    Lines 403-404: Consequently we changed here to: "The median hailstone size was 9 mm and the majority of hailstones were rather spherical with axis ratios greater 0.9."

9.  Figure 5(d) doesn't make sense until you get to Line 300. I understand why it makes sense to have with (a)-(c) as a single figure, but it may be worth noting the section it applies to in the caption.

    Now we added in the caption of Figure 5 that plot (d) belongs to Section 3.2.

10. Lines 306-309 might be more easily digested as a table, but that's more a preference.

    Here we would like to stay for now with the 3 sentences, unless it is more strictly recommended to use a table instead.

11. Section 4 – First paragraph could maybe go in intro, feels a little out of place here, but again more of a preference.

    In regard of the discussion of different techniques to identify hail stones in the subsequent paragraphs, we believe the first paragraph here does fit quite well, as the transparency of the hail stones is connected to the performance of the different techniques.

    We found that the last sentence here ("Thus, for hailstones with high transparency the approach used here might not work.") is sort of confusing and we now write for a better transition instead: "The effectiveness of various methods used to detect hailstones is influenced, in part, by the transparency of the ice."

12. Line 367 "as published in e.g. Knight (1986); Shedd et al. (2021)" -> "as published in Knight…"

Line 367: This has been adopted, "e.g." is deleted now.

13. Table 3 – Would it be possible to make this as a time series plot, T on left axis, RH on right (or similar)? It is more difficult to pick out the trends looking at a table. Noting the flight times as vertical lines or highlighted sections would further help in understanding the overall timeline of events.

Thank you for this suggestion and we think it is a good idea to put the temperature, information in a simple time series plot together with time information of the start of the hailfall (vertical black line) and drone flights (grey shaded vertical bars) to capture the soccer middle circle (as given in Table 2). Here is the version we suggest and we put into the manuscript:

[Figure]

Table 3 is then obsolete (all information is in the plot) and will be deleted.

14. Line 385 "what might effect" -> "which might affect"

Line 385: We changed the wording from "what might effect" to "which might affect".

Overall largest concern: Is this repeatable and generalizable? Your results are based on a single event used for training, validation and testing. Very interesting work though and sets the stage for future research that can begin to fine tune and hopefully more extensively validate these types of analyses.

As we describe in Table 4, the operational application of drone-based photogrammetry of hail is challenging and thus requires time and resources to build larger data sets incorporating several events. For looking into the broad applicability more data from hail events are needed. We hope that in a future, more researcher successfully execute such drone flights in the field and the community puts an effort in building up a database of aerial images. Such a database could help to further develop this approach and build more generalized models.

Other suggestions for potential future research:

(No expectation of these for this publication but curiosities that may be of interest to the authors.)

- Integration of thermal imagery. Even with low resolution, the integrated pixel values could provide useful information. By using the surface temp in areas with high probability and confidence of not having hail present as a background, you could use the differentials of other pixels to help include or exclude potential hailstones in conjunction with RGB techniques.

- Utilizing SfM result and applying R-CNN directly to mesh or point cloud rather than 2D orthomosaic. This would obviously require more computing power, but it would be interesting to see how it changes performance in 2D visually challenging environments (like taller grass).

Thank you very much for these two additional suggestions for future work and possible improvements of the method. We decided to take those ideas and include them in the outlook of the paper in Section 5:
Other ideas to test and potentially improve the techniques in the future are:
- Integration of thermal imagery to help exclude or include potential hailstones alongside the RGB image processing.
- Usage of SfM (Structure from Motion) results and application of Mask R-CNN directly to mesh or point clouds instead to the RGB orthophotos.